# Variance Driven Exploration: A Provable and Efficient Methodology for Pure Exploration in Highly Stochastic Environments

Khang Luong [1]   Nam Nguyen [1]   Hoang Ta [1]   Hung The Tran [2]   Tuan Dam [1]

## Abstract

We propose *Variance Driven Exploration* (VarDE), a principled approach for pure exploration in *highly stochastic environments*, where the exploration process is dominated by stochastic variance. VarDE is built on a fundamental principle: *sampling effort should be allocated to minimize the uncertainty of the final decision*. We formalize the uncertainty of the final decision through a smooth decision function and derive allocation rules that explicitly capture how stochastic noise in individual components affects the reliability of the final output. We apply this methodology to three core problems of pure exploration – Best Arm Identification (BAI), Monte Carlo Tree Search (MCTS), and Best-Policy Identification (BPI) – with theoretical guarantees on variance decay and simple regret. Empirically, we demonstrate consistent and significant improvements of VarDE over existing methods, with especially strong gains in highly stochastic environments.

## 1. Introduction

Many learning and planning problems in Reinforcement Learning reduce to a common goal: *making a reliable decision from noisy interactions*. In *pure exploration* settings, the agent is not judged by its performance during data collection but only by the final output—e.g., selecting the best arm in bandits, choosing the best root action in planning, or returning a near-optimal policy in reinforcement learning. Consequently, the central challenge is not balancing exploration and exploitation for cumulative reward but *allocating samples to minimize uncertainty in the final decision*.

[1]Hanoi University of Science and Technology, Hanoi, Vietnam [2]Quantum AI & Cyber Security Institute, FPT Corporation, Vietnam. Correspondence to: Tuan Dam <tuandq@soict.hust.edu.vn>.

*Proceedings of the 43rd International Conference on Machine Learning*, Seoul, South Korea. PMLR 306, 2026. Copyright 2026 by the author(s).

A large class of successful algorithms typically allocates samples using optimistic bounds. In bandits, fixed-budget best-arm identification methods allocate pulls via confidence bounds or elimination rules (Audibert et al., 2010; Gabillon et al., 2012; Wang et al., 2023). In planning, UCT-style Monte Carlo Tree Search (Kocsis & Szepesvári, 2006; Browne et al., 2012) uses optimism to prioritize promising actions. In episodic reinforcement learning, best-policy identification methods often rely on optimistic value estimates and confidence sets to guide exploration (Ménard et al., 2021). While these approaches are powerful, they share a structural limitation: they primarily regulate local estimation error of each stochastic component without explicitly quantifying how uncertainty in each component propagates to the final returned decision.

In highly stochastic environments, optimism-based rules may be dominated by stochastic fluctuations and mistake noise for evidence, overlooking the true signal. Under heteroscedastic variance, a sample is not equally informative across components: high-variance components may remain poorly estimated even after many pulls, while low-variance components quickly appear reliable. Table 1 gives a stylized three-arm snapshot that illustrates this failure mode. Arm 1 is the true best arm but is much noisier; after a number of pulls, its empirical mean can temporarily look worse than those of the two stable suboptimal arms. Empirical ranking may stop treating Arm 1 as competitive, even with a confidence bonus: $\hat{\mu}_1 + \frac{1}{\sqrt{n_1}} = 0.42 + \frac{1}{\sqrt{40}} < 0.58 < \hat{\mu}_2$, and thus stop sampling from it. The decision-relevant uncertainty, however, is concentrated in Arm 1, because additional samples from this arm are most likely to change the final recommendation. More generally, pure exploration decisions (e.g., an $\arg\max$ arm, a root action, or a greedy policy) are nonlinear functions of many estimates, so regulating *local* errors or bonuses does not necessarily reduce uncertainty of the *final* returned decision; this mismatch is amplified in planning and RL, where uncertainty propagates through value backups and can dominate the root choice. An effective pure exploration strategy should therefore ask a more direct question: *Which component, if sampled next, most reduces uncertainty in the final decision we will output?*

*Table 1.* A stylized heteroscedastic BAI snapshot. Arm 1 is optimal in expectation but has larger variance. Empirical ranking can favor the stable suboptimal arms, while decision-relevant uncertainty remains concentrated in Arm 1.

| Arm | Distribution | true $\mu_i$ | true $\sigma_i$ | $n$ sample | empirical $\hat{\mu}_i$ |
|-----|--------------|--------------|------------------|------------|--------------------------|
| **1** | Pr(X=2.0)=0.3, Pr(X=0.0)=0.7 | **0.60** | 0.92 | 40 | 0.42 |
| 2 | Pr(X=0.58)=0.5, Pr(X=0.60)=0.5 | 0.59 | 0.01 | 40 | **0.59** |
| 3 | Pr(X=0.55)=0.5, Pr(X=0.57)=0.5 | 0.56 | 0.01 | 40 | 0.56 |

We propose *Variance Driven Exploration* (VarDE), a general methodology that addresses this question by directly targeting decision-level uncertainty. VarDE formalizes the final decision as a (smooth) *decision function* of local empirical estimates, and uses a first-order sensitivity analysis to quantify each component's influence on the output. This yields a simple allocation principle: *sample where the (first-order) expected decrement in decision variance is largest*. The resulting rule is explicit, modular, and naturally balances two factors: (i) how strongly the final decision depends on a component (an *influence weight*), and (ii) how noisy that component is (its empirical variance).

We instantiate VarDE in three canonical pure exploration settings: **Best-Arm Identification** (BAI), **Monte Carlo Tree Search** (MCTS), and **Best-Policy Identification** (BPI). Across these settings, VarDE provides a unified lens on exploration: rather than focusing on local confidence bounds, it allocates effort to directly reduce uncertainty in the returned decision.

**Contributions.** Our main contributions are:

- **From decision-level variance analysis to a generic, implementable allocation rule.** In section 3, we introduce VarDE, a general methodology that treats the final recommendation as a smooth decision function of local empirical estimates, and derives an explicit *influence-weighted* decomposition of decision uncertainty (Lemma 3.4). From the variance decrement analysis (Proposition 3.6), we obtain a simple greedy rule that selects the component maximizing the expected reduction in the decision variance, combining influence weights with empirical variances.

- **Instantiations across bandits, planning, and RL.** In section 4, we derive concrete algorithms VARDE–BAI, VARDE–MCTS, and VARDE–Q-LEARNING, showing how influence weights and variance estimates can be computed in each setting.

- **Theory in stochastic regimes.** We provide a first-order constant characterization and a decision-variance decay guarantee (Theorem 3.7), together with correctness guarantees for the final recommendation: the misidentification probability decays exponentially in BAI (Theorem 4.5) and MCTS (Corollary 4.8), and VARDE–Q-LEARNING converges to an optimal policy with probability 1 (Theorem 4.11).

- **Empirical validation in stochastic environments.** Section 5 demonstrates consistent empirical improvements over strong baselines across bandits, planning, and RL benchmarks, with the largest gains in highly stochastic environments.

## 2. Related Work

**Best-arm identification (BAI).** Fixed-budget BAI methods can be broadly grouped by how they control uncertainty. A first family allocates pulls using *confidence intervals* (CI)-derived indices: classic examples include UCB-E (Audibert et al., 2010) and more refined gap-based rules such as UGAPE (Gabillon et al., 2012), which use optimistic estimates to prioritize arms that are plausibly optimal. A second family follows a *successive reject / elimination* principle, progressively discarding arms and focusing the remaining budget on a shrinking candidate set; this includes SUCCESSIVE HALVING (Karnin et al., 2013) and SUCCESSIVE REJECTS (Audibert et al., 2010). More recently, CONTINUOUS REJECTS (Wang et al., 2023) refines this reject-based view via continuous-time/large-deviation-inspired allocation.

While both CI-based and reject-based designs are effective, they are largely driven by mean-based evidence (gaps, confidence radii, or reject schedules). They do not explicitly model or optimize how heteroscedastic reward variance influences the reliability of the final decision. In contrast, VARDE–BAI is explicitly variance-aware: it selects the next arm by maximizing a *decision-variance decrement* of the form (influence)×(variance), prioritizing samples that most reduce uncertainty in the final recommendation.

**Monte Carlo Tree Search (MCTS).** A standard approach to exploration in tree search is UCT (Kocsis & Szepesvári, 2006), which extends bandit optimism to internal nodes via UCB-style bonuses to balance exploitation of high-value actions and exploration of uncertain ones. In highly stochastic domains, several alternatives encourage broader search by injecting randomness or regularization into action selection, such as maximum-entropy planning (MENTS) (Xiao et al., 2019), convex-regularized variants (RENTS/TENTS/DENTS) (Dam et al., 2021), and Boltzmann exploration in trees (BTS) (Painter et al., 2023).

These methods diversify exploration through optimism or entropy mechanisms, but they do not explicitly optimize how uncertainty along different edges influence the *action recommendation*. VARDE–MCTS differs by explicitly weighting edges by both their influence on the action decision and their empirical return variance.

**Best-policy identification (BPI) in RL.** Pure-exploration RL (and BPI in particular) has been studied from minimax and adaptive perspectives, with algorithms that provide strong finite-sample guarantees by explicitly reasoning over model uncertainty (Ménard et al., 2021; Domingues et al., 2021; Marjani & Proutiere, 2021). Many of the best-theory approaches are *model-based*: they maintain confidence sets for rewards/transitions and repeatedly solve (optimistic) planning problems to decide where to sample next (Azar et al., 2017; Al Marjani et al., 2021). While statistically powerful, these methods are often implementation-heavy and computationally demanding due to repeated dynamic programming / optimistic MDP solving and the need to manipulate confidence sets over $(r, P)$, which can scale poorly with $|\mathcal{S}|$ and $|\mathcal{A}|$ (Ménard et al., 2021; Al Marjani et al., 2021). In contrast, *model-free* baselines such as Q-LEARNING (and $\varepsilon$-greedy variants), Q-UCB (Jin et al., 2018), PSRL (Osband et al., 2013), and recent fixed-budget model-free BPI methods (Russo & Proutiere, 2023) are comparatively simple and scalable, but can be brittle in highly stochastic environments where naive exploration is inefficient. VARDE is complementary: it provides a lightweight, plug-in sampling rule that targets uncertainty of the *final greedy decision* via a smooth surrogate and an empirical variance signal (TD-target variance), yielding a practical decision-aware alternative without requiring full model construction or repeated optimistic planning.

## 3. Variance Driven Exploration

### 3.1. Preliminaries and evaluation

Throughout this paper, $[n] = \{1, 2, \ldots, n\}$, $\|\cdot\|_2$ denotes the Euclidean norm, $\|\cdot\|_{\mathrm{op}}$ denotes the operator norm, and $\|\cdot\|_\infty$ denotes the $\ell^\infty$ norm: $\|x\|_\infty := \max_i |x_i|$.

We formalize the principle of **VarDE** as a general statistical methodology for decision-level uncertainty minimization. Let $X_1, \ldots, X_n$ denote local stochastic components (arms, leaf values, or state–action returns), each with unknown mean $\mu_i$ and variance $\sigma_i^2 < \infty$; for each component $i \in [n]$, we can collect i.i.d. samples to form the empirical mean $\hat{\mu}_i$. Let $Y = f(\hat{\mu}_1, \ldots, \hat{\mu}_n)$ denote a global scalar *decision variable* depending on the empirical means $\hat{\mu}$, such as the root value in a planning tree or a smooth surrogate of the best-arm decision in BAI.

Our goal in this section is to derive a simple, unified sampling rule that greedily reduces the decision-level uncertainty $\mathrm{Var}[Y]$ by combining (i) how strongly each component influences $Y$ and (ii) how noisy that component is.

**Roadmap.** To make this principle actionable, we first relate $\mathrm{Var}[Y]$ to the component-wise estimation errors of $\hat{\mu}$.

Assumption 3.1 allows a Taylor linearization of $f$ on the bounded empirical domain, which yields *influence weights* $w_i(\mu)$. We then decompose $\mathrm{Var}[Y]$ into an influence-weighted sum of local variances and derive a one-sample *decision-variance decrement*, leading to the greedy VarDE sampling rule. Proofs are deferred to Appendix A.

**Assumption 3.1** (VarDE regularity conditions). *Assume observations for every component lie in a common bounded interval $[a, b]$; hence $\mu$ and every empirical vector $\hat{\mu}(t)$ lie in the compact convex domain $D := [a, b]^n$. Assume the decision function $f : \mathbb{R}^n \to \mathbb{R}$ is twice continuously differentiable on an open convex set containing $D$, and that its Hessian is uniformly bounded and Lipschitz on $D$: there exist constants $M, L < \infty$ such that, for all $x, y \in D$,*

$$\|H_f(x)\|_{\mathrm{op}} \leq M, \qquad \|H_f(x) - H_f(y)\|_{\mathrm{op}} \leq L\|x - y\|_2.$$

### 3.2. Influence Weights

**Lemma 3.2** (Local linear approximation). *Under Assumption 3.1, for any $\hat{\mu} \in D = [a, b]^n$,*

$$f(\hat{\mu}) = f(\mu) + \nabla f(\mu)^\top (\hat{\mu} - \mu) + \mathcal{O}\left(M\|\hat{\mu} - \mu\|_2^2\right).$$

Neglecting the quadratic remainder in Lemma 3.2 yields the first-order approximation

$$Y = f(\hat{\mu}) \approx c + \sum_{i=1}^n w_i(\mu)\, \hat{\mu}_i,$$

where $w(\mu) := \nabla f(\mu)$, and $c = f(\mu) - w(\mu)^\top \mu$.

We call $w_i(\mu)$ the *influence weights*, as they quantify how sensitive the global decision variable $Y = f(\hat{\mu})$ is to small perturbations in the local estimate $\hat{\mu}_i$. We use the following nondegeneracy assumption.

**Assumption 3.3** (Nonvanishing influence weights). *There exists $\rho > 0$ such that*

$$|w_i(x)| \geq \rho, \qquad \forall x \in D, \ \forall i \in [n]. \tag{3.1}$$

### 3.3. Variance Analysis

Using influence weights, we can decompose the variance of the decision variable:

**Lemma 3.4** (First-order variance decomposition). *Under Assumption 3.1, the variance floor, Assumption 3.3, and independence across components, let $Y_t = f(\hat{\mu}(t))$ denote the decision variable after $t$ total samples, then*

$$\mathrm{Var}[Y_t] = \sum_{i=1}^n w_i(\mu)^2 \frac{\sigma_i^2}{N_i(t)} + O(t^{-2}).$$

Decision uncertainty $\mathrm{Var}[Y_t]$ is not simply the sum of local estimation variances. Each local variance is scaled by the squared influence of that component on the final decision variable.

**Lemma 3.5** (One-Sample Variance Decrement)**.** *For each $i \in [n]$, when one additional sample is collected for $X_i$, the change in its local variance is:*

$$\Delta \operatorname{Var}[\hat{\mu}_i] = -\frac{\sigma_i^2}{N_i(N_i + 1)}.$$

Combining the first-order variance decomposition with the local decrement yields the first-order global variance update of VarDE.

**Proposition 3.6** (Global variance decrement)**.** *Fix a step $t$ in which the algorithm collects one additional sample from component $i_t \in [n]$. Under Assumption 3.1, the variance floor, Assumption 3.3, and independence across components,*

$$\operatorname{Var}[Y_{t+1}] - \operatorname{Var}[Y_t] = -w_{i_t}(\mu)^2 \frac{\sigma_{i_t}^2}{N_{i_t}(t)(N_{i_t}(t) + 1)} + O(t^{-2}).$$

This equation expresses how a single new sample from component $i_t$ reduces the decision-level variance in proportion to its influence weight $w_{i_t}^2$ and its local noise $\sigma_{i_t}^2$.

### 3.4. Optimal Sampling Rule

Since each additional sample yields the marginal variance decrement above, the optimal greedy sampling rule is to select the component that maximizes the (first-order) variance reduction:

$$\tilde{i} \in \arg\max_{i \in [n]} w_i(\hat{\mu})^2 \frac{\tilde{\sigma}_i^2}{N_i(N_i + 1)},$$

where $\tilde{\sigma}_i^2 = \max(\hat{\sigma}_i^2, \bar{\sigma}^2)$ and $\bar{\sigma}^2 > 0$ is the variance floor. This formulation highlights VarDE as a methodology that unifies stochastic decision processes–from flat bandits to hierarchical planners and value-based learners–under a single principle of minimizing decision-level uncertainty.

With this sampling rule, we can establish the following decision-level variance decay guarantee.

**Theorem 3.7** (Variance decay of VarDE)**.** *Under Assumption 3.1, the variance floor, Assumption 3.3, and independence across components, VarDE satisfies $N_i(t) = \Omega(t)$ for every component and therefore*

$$\operatorname{Var}[Y_t] = \mathcal{O}(t^{-1}).$$

**Remark 3.8** (First-order allocation constant)**.** *Theorem 3.7 gives the variance-decay exponent. The leading constant is the quantity that distinguishes well-spread allocations. Appendix A.7 shows that if $N_i(T) = p_i T + O(1)$ with $p_i > 0$ and $\sum_i p_i = 1$, then*

$$\operatorname{Var}[Y_T] = \frac{1}{T}C(p) + O(T^{-2}), \qquad C(p) = \sum_{i=1}^{n} \frac{w_i(\mu)^2 \sigma_i^2}{p_i}.$$

*The unique minimizer of this first-order constant is $p_i^\star \propto |w_i(\mu)|\sigma_i$, giving $C_\star = (\sum_i |w_i(\mu)|\sigma_i)^2$. Thus VarDE is not claiming a better exponent than all well-spread allocations; its asymptotic target is the optimal first-order constant induced by the decision-level variance decomposition.*

## 4. Applications

We now instantiate VarDE in three canonical pure-exploration settings–Best-Arm Identification (BAI), Monte Carlo Tree Search (MCTS), and Best-Policy Identification (BPI)–showing how a common decision-level uncertainty objective yields simple sampling rules and corresponding theoretical guarantees. All proofs are deferred to Appendices B, C, and D.

### 4.1. Best-Arm Identification Bandit

#### 4.1.1. PROBLEM SETTING

We consider $K$ stochastic arms $\{X_i\}_{i=1}^K$ with unknown means $\mu_i = \mathbb{E}[X_i]$ and finite variances $\sigma_i^2 = \operatorname{Var}[X_i]$. At each round $t$, the learner selects an arm $i_t$ and observes a reward $r_t \sim X_{\tilde{i}_t}$, with $r_t \in [0, 1]$. The objective is to identify the best arm $i^* = \arg\max_{i \in [K]} \mu_i$. Accordingly, we seek to minimize the *simple regret*

$$\mathcal{R}_T = \mathbb{E}[\mu_{i^*} - \mu_{\hat{i}_T}],$$

where $\hat{i}_T$ denotes the arm recommended after $T$ samples.

The decision value in BAI is the maximum empirical mean. Instead of the non-differentiable $\max_i \hat{\mu}_i$, we adopt LogSumExp (LSE) as a smooth approximation:

$$Y_\tau = \operatorname{LSE}_\tau(\hat{\mu}) = \tau \log \sum_{i=1}^{K} e^{\hat{\mu}_i/\tau},$$

where $\tau > 0$ is a temperature parameter.

To apply VarDE, we require a smooth surrogate of $\max_i \hat{\mu}_i$ with controlled curvature. The following lemmas show that $\operatorname{LSE}_\tau$ (i) approximates $\max$ within $\tau \log K$, (ii) has a uniformly bounded Hessian, and (iii) has Lipschitz Hessian on bounded domains, thereby verifying the smoothness requirements of Assumption 3.1 for the BAI decision surrogate.

**Lemma 4.1** (LSE bound)**.** *For any vector $\hat{\mu} \in \mathbb{R}^K$ and any $\tau > 0$,*

$$\max_i \hat{\mu}_i \leq \operatorname{LSE}_\tau(\hat{\mu}) \leq \max_i \hat{\mu}_i + \tau \log K.$$

**Lemma 4.2** (LSE curvature)**.** *For any fixed $\tau > 0$ and any $\hat{\mu} \in \mathbb{R}^K$, the Hessian of $\operatorname{LSE}_\tau$ satisfies*

$$\left\| \nabla^2 \operatorname{LSE}_\tau(\hat{\mu}) \right\|_{\mathrm{op}} \leq \frac{1}{2\tau}.$$

**Lemma 4.3** (LSE Hessian Lipschitzness). *For any fixed $\tau > 0$, $\nabla^2 \mathrm{LSE}_\tau$ is Lipschitz on any compact convex domain $D \subset \mathbb{R}^K$. More explicitly, there exists $L_{\tau,K} < \infty$ such that, for all $x, y \in D$,*

$$\left\| \nabla^2 \mathrm{LSE}_\tau(x) - \nabla^2 \mathrm{LSE}_\tau(y) \right\|_{\mathrm{op}} \leq L_{\tau,K} \|x - y\|_2.$$

*One may take $L_{\tau,K} = 3K^{3/2}/\tau^2$.*

The temperature $\tau$ therefore controls a bias–smoothness tradeoff. By Lemma 4.1, smaller $\tau$ makes $\mathrm{LSE}_\tau$ a tighter approximation to the hard maximum, with bias at most $\tau \log K$. At the same time, Lemmas 4.2 and 4.3 show that the curvature and Hessian-Lipschitz constants scale as $O(\tau^{-1})$ and $O(\tau^{-2})$, respectively. Thus very small $\tau$ gives a more faithful decision surrogate but amplifies the higher-order Taylor remainder in the variance approximation, whereas larger $\tau$ smooths the decision variable and stabilizes the first-order variance surrogate at the cost of additional approximation bias.

### 4.1.2. VARDE–BAI ALGORITHM

We now instantiate VarDE in the fixed-budget best-arm identification setting. With the smooth decision variable $Y_\tau = \mathrm{LSE}_\tau(\hat{\mu})$, VarDE reduces to allocating pulls across arms to greedily decrease the decision-level variance $\mathrm{Var}[Y_\tau]$.

Under the $\mathrm{LSE}_\tau$ decision variable, the **influence weight** of each arm is

$$w_i(\hat{\mu}) = \frac{\partial Y_\tau}{\partial \hat{\mu}_i} = \frac{e^{\hat{\mu}_i/\tau}}{\sum_{j=1}^K e^{\hat{\mu}_j/\tau}}.$$

**Lemma 4.4** (LSE nonvanishing influence weights). *Fix $\tau > 0$ and a compact box $D = [a, b]^K$. For every $x \in D$ and every $i \in \{1, \ldots, K\}$,*

$$\frac{1}{K} e^{(a-b)/\tau} \leq w_i(x) \leq 1.$$

*In particular, the LSE influence weights verify Assumption 3.3 on $D$ with $\rho = K^{-1}e^{(a-b)/\tau} > 0$.*

VARDE–BAI greedily selects the arm that can contribute the largest expected variance decrement, applying the optimal sampling rule in Subsection 3.4 and updating $\hat{\mu}, \hat{\sigma}^2, N$ online. The complete algorithm is deferred to Appendix E (Algorithm 1).

### 4.1.3. THEORETICAL GUARANTEES

We next state finite-sample guarantees for VARDE–BAI in the fixed-budget setting. Under standard bounded-reward assumptions and a unique best arm, we show that VARDE–BAI identifies the optimal arm with exponentially small error probability, which in turn implies an exponential decay of the simple regret.

**Theorem 4.5** (VARDE–BAI error probability). *Assume rewards are supported on $[0, 1]$ and that the best arm $i^*$ is unique.* VARDE–BAI *(Alg. 1) with temperature $\tau > 0$ and warm start $\eta > 0$ satisfies*

$$\Pr\{\hat{i}_T \neq i^*\} \leq C \exp(-cT),$$

*for some constants $C, c > 0$ that may depend on $K$, $\tau$, $\bar{\sigma}$, and the gaps $\Delta_i = \mu_{i^*} - \mu_i$.*

**Corollary 4.6** (VARDE–BAI simple regret). *Under the assumptions of Theorem 4.5, the simple regret of* VARDE–BAI *satisfies*

$$\mathcal{R}_T \leq \Delta_{\max} \Pr\{\hat{i}_T \neq i^*\} = \exp(-\Omega(T)),$$

*where $\Delta_{\max} := \max_{i \neq i^*}(\mu_{i^*} - \mu_i)$.*

This exponential decay of the simple regret is order-optimal and matches the minimax lower bound in fixed-budget BAI up to constant factors (Audibert et al., 2010).

**Other Decision Functions.** While we focus on the $\mathrm{LSE}_\tau$ decision function for simplicity, VarDE can be adapted to other smooth approximations of the maximum function with analogous guarantees under similar regularity conditions.

## 4.2. Monte Carlo Tree Search

### 4.2.1. PROBLEM SETTING

We plan in an episodic Markov Decision Process (MDP) with horizon $H$, discount factor $\gamma \in (0, 1]$, finite state set $\mathcal{S}$ and action set $\mathcal{A}$. We assume access to a forward model that can simulate transitions and rewards along a trajectory. Given a state $s$ and action $a \in \mathcal{A}(s)$ (the set of feasible actions at $s$), a call to the model returns a next state $s' \sim P(\cdot \mid s, a)$ and a reward sample $\hat{r} \sim \nu(\cdot \mid s, a)$ supported on $[0, 1]$, with mean $r(s, a)$; the next call proceeds from $s'$.

The optimal action-value function is

$$Q^*(s, a) = \sup_\pi \mathbb{E}^\pi \left[ \sum_{t=0}^{H-1} \gamma^t r(s_t, a_t) \,\middle|\, s_0 = s, a_0 = a \right],$$

where the supremum is over deterministic policies $\pi$, and the expectation is over trajectories $s_0, a_0, s_1, \ldots, s_H$ with $a_t = \pi(s_t)$ and $s_{t+1} \sim P(\cdot \mid s_t, a_t)$.

Our objective is to identify the best root action $a^* = \arg\max_{a \in \mathcal{A}(s_0)} Q^*(s_0, a)$ using $T$ simulated trajectories. Accordingly, we minimize the simple regret

$$\mathcal{R}_T = \mathbb{E}[Q^*(s_0, a^*) - Q^*(s_0, \hat{a}_T)],$$

where $\hat{a}_T$ is the action recommended after $T$ simulations.

### 4.2.2. VARDE–MCTS ALGORITHM

To apply VarDE in MCTS, we view each state–action pair $(s, a)$ as a local stochastic component whose return is estimated by the empirical $\hat{Q}(s, a)$. The global decision at a state $s$ is the recommended action $\hat{a}$, which depends on comparing action values through the maximization $V^*(s) = \max_{a \in \mathcal{A}(s)} Q^*(s, a)$. Since max is non-differentiable, we replace it with a smooth surrogate so that influence weights and variance decrements can be defined in closed form. We therefore use a smooth surrogate of $\max_{a \in \mathcal{A}(s)} \hat{Q}(s, a)$ via $\mathrm{LSE}_\tau$:

$$Y_\tau(s) = \mathrm{LSE}_\tau(\hat{Q}(s, \cdot)) = \tau \log \sum_{a \in \mathcal{A}(s)} e^{\hat{Q}(s,a)/\tau},$$

where $\hat{Q}(s, a)$ is the empirical value of action $a$ at state $s$. The influence weight of action $a$ at state $s$ is

$$w(s, a) = \frac{\partial Y_\tau(s)}{\partial \hat{Q}(s, a)} = \frac{e^{\hat{Q}(s,a)/\tau}}{\sum_{b \in \mathcal{A}(s)} e^{\hat{Q}(s,b)/\tau}}.$$

The uncertainty of action $a$ at state $s$ is evaluated by the empirical variance $\hat{\sigma}(s, a)$ of cumulative returns $\hat{G}(s, a)$ when taking $a$ at $s$.

$$\hat{G}(s, a) := \sum_{t=0}^{H-1} \gamma^t \hat{r}(s_t, a_t),$$
$$(s_0, a_0) = (s, a), \ s_{t+1} \sim P(\cdot \mid s_t, a_t).$$

During selection phase, VARDE–MCTS greedily chooses the action that yields the largest estimated (first-order) variance decrement, applying the optimal sampling rule in Subsection 3.4. The simulation phase and backpropagation phase follow standard MCTS procedures with max backup. The complete algorithm is deferred to Appendix E (Algorithm 2).

### 4.2.3. THEORETICAL GUARANTEES

We next provide concentration guarantees for VARDE–MCTS: value estimates concentrate exponentially fast in the number of simulations, which implies an exponentially small probability of recommending a suboptimal root action.

**Theorem 4.7** (VARDE–MCTS value estimates). *Consider a VARDE–MCTS process. For any depth $t \in \{0, \ldots, H\}$, any node $s_t$, and any fixed tolerance $\varepsilon > 0$, there exist constants $C_{s_t,\varepsilon} > 0$ and $k_{s_t,\varepsilon} > 0$ such that, conditionally on $N(s_t) = n \geq 1$,*

$$\Pr\left(\left|\hat{V}(s_t) - V^*(s_t)\right| > \varepsilon\right) \leq C_{s_t,\varepsilon} \exp\left(-k_{s_t,\varepsilon}\, \varepsilon^2 n\right).$$

*Moreover, at the root node $s_0$, after $T$ simulations, there exist constants $C_\varepsilon > 0$ and $k_\varepsilon > 0$ such that*

$$\Pr\left(\left|\hat{V}(s_0) - V^*(s_0)\right| > \varepsilon\right) \leq C_\varepsilon \exp\left(-k_\varepsilon\, \varepsilon^2 T\right).$$

**Corollary 4.8** (VARDE–MCTS error probability). *Under the assumptions of Theorem 4.7, the error probability of VARDE–MCTS satisfies*

$$\Pr\{\hat{a}_T \neq a^*\} \leq \exp\left(-\Omega(T)\right).$$

This exponential decay of the error probability matches the standard exponential concentration rates established for Monte Carlo Tree Search in stochastic environments (Browne et al., 2012; Dam et al., 2021; Painter et al., 2023).

### 4.3. Best Policy Identification

#### 4.3.1. PROBLEM SETTING

We consider the non-episodic infinite-horizon case under the same MDP and forward model assumptions as in Section 4.2.1. When learning under the infinite-horizon MDP, we also need the following assumption.

**Assumption 4.9** (Communicating MDP). *For any states $s, s' \in \mathcal{S}$, there exists a stationary policy $\pi$ such that the expected hitting time of $s'$ starting from $s$ is finite:*

$$\mathbb{E}_s^\pi[T_{s'}] < \infty, \qquad T_{s'} := \inf\{t \geq 0 : s_t = s'\}.$$

Given a sampling budget of $T$ steps, the goal of *best-policy identification* (BPI) is to output a policy $\hat{\pi}$ with small *simple regret*

$$\mathcal{R}_T = V^*(s_0) - V^{\hat{\pi}}(s_0),$$

where

$$V^{\hat{\pi}}(s) = \mathbb{E}^{\hat{\pi}}\left[\sum_{t=0}^{\infty} \gamma^t r(s_t, a_t) \,\middle|\, s_0 = s, a_t = \hat{\pi}(s_t)\right]$$

is the value of policy $\hat{\pi}$ at state $s$.

#### 4.3.2. VARDE–Q-LEARNING ALGORITHM

We apply the VarDE sampling rule to Q-learning, using a decision function and influence weights defined analogously to VARDE–MCTS. However, instead of targeting the variance of the cumulative return, which cannot be computed in non-episodic infinite-horizon MDP, we use the variance of the one-step TD target

$$y = \hat{r} + \gamma \max_{a \in \mathcal{A}} \hat{Q}(s', a),$$

which captures *environmental stochasticity* (reward and transition randomness) as well as bootstrapping noise. The complete VARDE–Q-LEARNING algorithm is deferred to Appendix E (Algorithm 3).

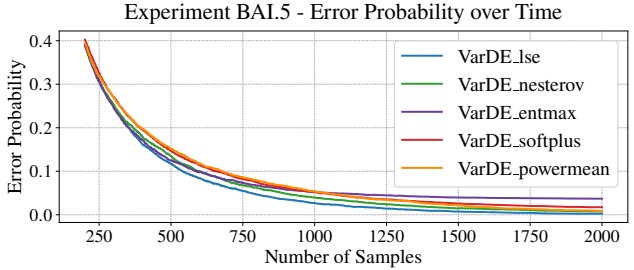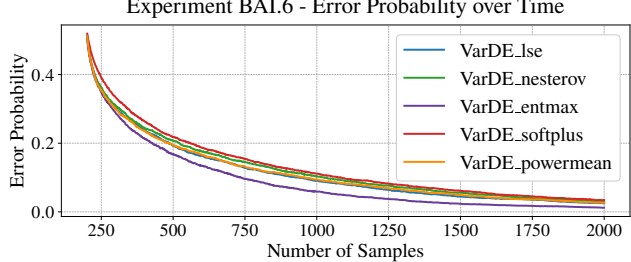

*Figure 1.* Error probability over time of VARDE–BAI with different smooth decision functions (Appendix F.1 gives function definitions and hyperparameters).

### 4.3.3. THEORETICAL GUARANTEES

We state guarantees in the tabular discounted setting. At a high level, we separate two ingredients: (i) a *coverage* condition ensuring sufficient exploration of all state–action pairs, under which tabular Q-learning converges; and (ii) a standard performance-loss bound translating $Q$-function error into policy suboptimality.

**Lemma 4.10** (Coverage). *Assume rewards satisfy $r(s, a) \in [0, 1]$, $|\mathcal{S}|, |\mathcal{A}| < \infty$, and $\gamma \in (0, 1)$. Let $N_T(s, a)$ be the number of visits to $(s, a)$ up to time $T$. Then every $(s, a)$ is visited infinitely often by* VARDE–Q-LEARNING, *i.e.,*

$$N_T(s, a) \to \infty \quad as \ T \to \infty, \ \forall (s, a) \in \mathcal{S} \times \mathcal{A}.$$

Under diminishing step sizes satisfying the standard Robbins–Monro conditions (as used in Alg. 3) and infinite visitation, tabular Q-learning converges to the optimal action-value function. According to Watkins & Dayan (1992), we have the following convergence guarantees.

**Theorem 4.11** (VARDE–Q-LEARNING convergence). *Under the same assumptions as Lemma 4.10, Algorithm 3 satisfies*

$$\|Q_T - Q^*\|_\infty \xrightarrow[T \to \infty]{} 0 \qquad with \ probability \ 1.$$

*Consequently, the greedy policies $\hat{\pi}_T$ are optimal for all sufficiently large $T$ (up to tie-breaking).*

This result is an asymptotic consistency guarantee. Unlike the BAI and MCTS guarantees above, it is not a non-asymptotic fixed-budget BPI bound; deriving such a bound for the adaptive variance-driven Q-learning rule remains an important theoretical direction.

**Lemma 4.12** (Greedy policy suboptimality from $Q$-error). *Let $\hat{\pi}$ be greedy w.r.t. $\hat{Q}$. If $\|\hat{Q} - Q^*\|_\infty \le \varepsilon$, then*

$$\|V^{\hat{\pi}} - V^*\|_\infty \le \frac{2\varepsilon}{1 - \gamma}.$$

*In particular, $\mathcal{R}_T \le \frac{2}{1-\gamma}\|Q_T - Q^*\|_\infty$.*

*Table 2.* Final error probability (in %) for Best-Arm Identification under four benchmark settings.

| SETTING NO. OF PULLS | BAI.1 1200 | BAI.2 1000 | BAI.3 200 | BAI.4 150 |
|---|---|---|---|---|
| UNIFORM | 33.23 | 38.69 | 28.53 | 32.98 |
| SH | 29.05 | 29.08 | 15.75 | 19.56 |
| SR | 16.06 | 20.70 | 12.39 | 15.41 |
| CR-A | 17.00 | 21.07 | 9.83 | 11.93 |
| CR-C | 16.71 | 20.84 | 11.80 | 12.71 |
| UCBE$_2$ | 15.67 | 21.59 | 12.93 | 19.32 |
| UCBE$_4$ | 19.38 | 25.92 | 16.56 | 22.99 |
| UCBE$_8$ | 23.54 | 28.59 | 19.25 | 26.09 |
| UGAPE$_2$ | 15.64 | 21.75 | 13.48 | 20.55 |
| UGAPE$_4$ | 20.43 | 26.43 | 17.50 | 23.48 |
| UGAPE$_8$ | 24.96 | 30.40 | 19.55 | 26.01 |
| VARDE$_{0.05}$ | **12.87** | **17.27** | 11.56 | **11.83** |
| VARDE$_{0.1}$ | 14.04 | 19.31 | **7.34** | 13.81 |
| VARDE$_{0.15}$ | 16.56 | 21.26 | 8.21 | 18.52 |

## 5. Experiments

**Common protocol.** Unless otherwise stated, curves report the mean over independent runs and shaded regions indicate 95% confidence intervals. Hyperparameters for each baseline are tuned by grid search on a separate validation split (same budget as evaluation), and we report the best validation configuration. Implementation details and additional results are provided in Appendix F. Code is available at https://github.com/luongkhang04/VarDE.

### 5.1. Best-Arm Identification

We evaluate VARDE–BAI on four standard Best-Arm Identification (BAI) benchmarks and compare against UNI-FORM, UCB-E (Audibert et al., 2010) ($\alpha \in \{2, 4, 8\}$), UGAPE (Gabillon et al., 2012) ($\alpha \in \{2, 4, 8\}$), SUCCESSIVE HALVING (Karnin et al., 2013) (SH), SUCCESSIVE REJECTS (Audibert et al., 2010) (SR), and CONTINUOUS REJECTS (Wang et al., 2023) (CR). All results report the error probability under a fixed sampling budget, averaged over 20,000 independent runs. Full experimental details are deferred to Appendix F.1.

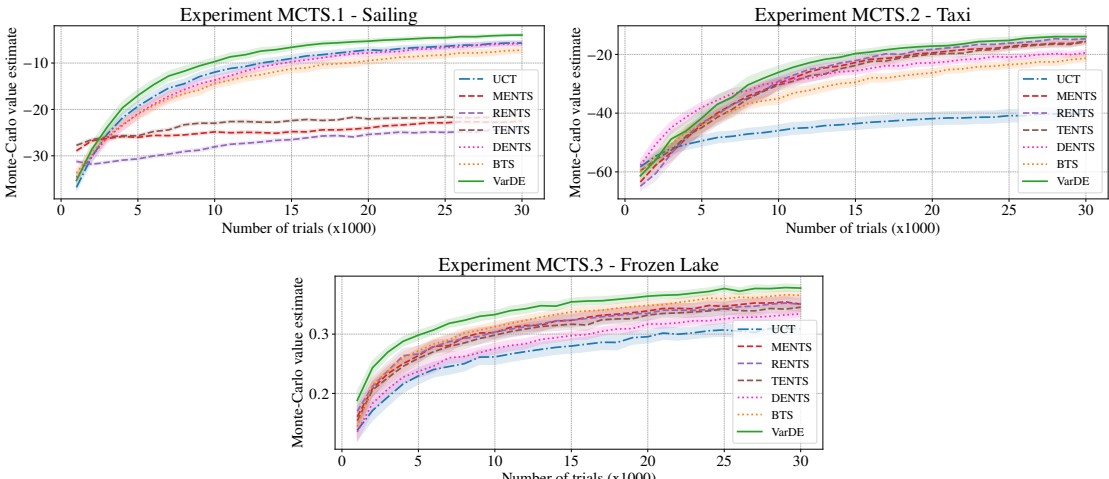

*Figure 2.* Monte Carlo value estimates of the recommended root action/policy during planning (mean $\pm$ 95% CI over runs).

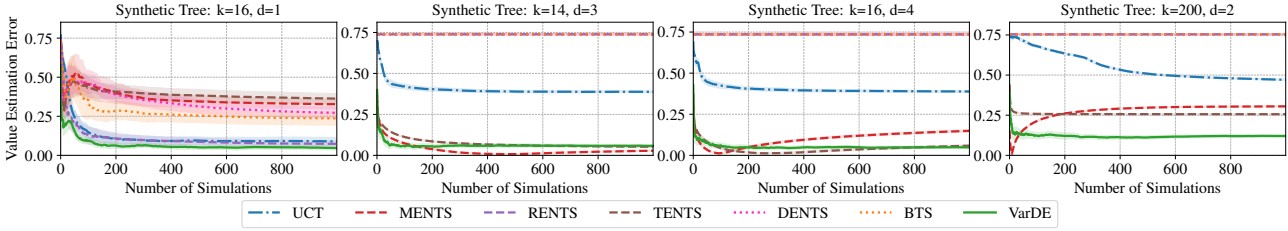

*Figure 3.* Value estimation error of VARDE–MCTS and other algorithms on synthetic tree (Lower is better).

Table 2 summarizes the results. VARDE–BAI achieves the lowest error probability in all four experiments. These results empirically validate the advantage of our method.

We also provide empirical comparisons of VARDE–BAI with different decision functions. Details about the implemented decision functions are provided in Appendix F.1. The results, presented in Figure 1, show that VARDE–BAI with various smooth decision functions performs competitively, demonstrating the flexibility of the VarDE methodology.

Appendix F.1.1 reports additional ablations and parameter sensitivity studies. The ablations show that influence weights and empirical variance are both necessary: combining the two consistently outperforms using either component alone. The sensitivity results show that the method is more sensitive to $\tau$ than to $\bar{\sigma}$. The variance floor mainly stabilizes early estimates, performance near the best $\tau$ is comparatively stable across a broad range of $\bar{\sigma}$. Appendix F.1.2 directly quantifies this approximation error by comparing the first-order variance surrogate with the full nonlinear variance of the LSE decision function. The discrepancy is largest in the sharp small-temperature regime and becomes substantially smaller for moderate $\tau$ as sampling progresses.

## 5.2. Monte Carlo Tree Search

We evaluate VARDE–MCTS on three grid-world environments (SAILING, TAXI, FROZENLAKE) and a synthetic tree benchmark, and compare against UCT (Kocsis & Szepesvári, 2006), MENTS (Xiao et al., 2019), RENTS, TENTS (Dam et al., 2021), DENTS, and BTS (Painter et al., 2023). For the grid worlds we report Monte Carlo value estimates of the recommended root action/policy after every 1000 trials (simulations), averaged over 100 independent runs (Fig. 2). For the synthetic tree we report root value estimation error over simulations (Fig. 3). Full experimental details are deferred to Appendix F.2.

Across all environments with sparse rewards (FROZENLAKE) or dense stochastic rewards (SAILING), the policies recommended by VARDE–MCTS achieve the highest Monte Carlo value estimates throughout planning. On the synthetic tree benchmark, VARDE–MCTS attains the lowest root value estimation error across a range of branching factors and depths. Overall, these results highlight the advantage of decision-variance minimization in MCTS, particularly in highly stochastic settings (SAILING and SYNTHETIC TREE with $k = 200, d = 2$), where several baselines exhibit unstable search and fail to reliably reduce the root value estimation error.

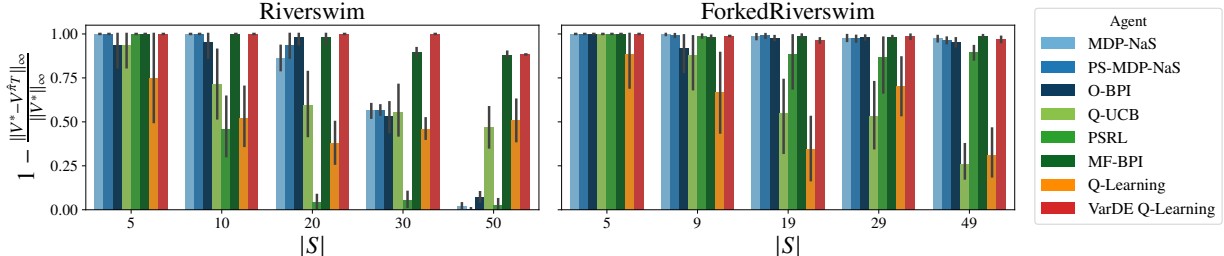

*Figure 4.* Normalized value proximity of VARDE–Q-LEARNING against model-free and model-based methods after a fixed interaction budget $T$ on RIVERSWIM and FORKEDRIVERSWIM (mean $\pm$ 95% CI).

### 5.3. Best Policy Identification

We evaluate VARDE–Q-LEARNING on RIVERSWIM and FORKEDRIVERSWIM across a range of state sizes $|\mathcal{S}|$. Each method interacts with the environment for a fixed budget of $T$ steps and returns the greedy policy $\hat{\pi}_T$ induced by its final $Q$-estimate. We evaluate $\hat{\pi}_T$ *exactly* on the true MDP (iterative policy evaluation with tolerance $10^{-6}$) and compute $V^*$ by policy iteration. In the main plots we report the normalized value proximity $1 - \|V^* - V^{\hat{\pi}_T}\|_\infty / \|V^*\|_\infty$. Baselines include model-free Q-UCB (Jin et al., 2018), PSRL (Osband et al., 2013), MF-BPI (Russo & Proutiere, 2023), and $\varepsilon$-Q-LEARNING, as well as model-based MDP-NAS, its posterior-sampling variant PS-MDP-NAS (Al Marjani et al., 2021), and O-BPI (Russo & Proutiere, 2023). Full experimental details and applied hyperparameters are provided in Appendix F.3.

Fig. 4 demonstrates that VARDE–Q-LEARNING is substantially more robust than existing baselines as the environments scale. On RIVERSWIM, multiple baselines deteriorate rapidly with increasing $|S|$ and show large run-to-run variability, consistent with the difficulty of learning from rare and noisy returns. In contrast, VARDE–Q-LEARNING stays near-optimal across all tested sizes and has tight confidence intervals, indicating both higher final policy quality and greater stability under stochasticity. A similar pattern holds on FORKEDRIVERSWIM: VARDE–Q-LEARNING matches or exceeds the best-performing baselines across sizes and remains consistently stable. These results support the main premise of VarDE—minimizing decision-level uncertainty yields superior fixed-budget performance in highly stochastic environments.

## 6. Discussion and Future Work

VarDE casts pure exploration as *decision-variance minimization*: allocate samples to the component that is both (i) influential to the final recommendation and (ii) empirically noisy. Across BAI, MCTS, and best-policy identification, this view is most beneficial in highly stochastic and heteroscedastic settings, where common optimism/entropy heuristics may misallocate budget by reacting to noise rather than to decision-relevant uncertainty.

**Limitations.** Our rule is derived from a first-order variance approximation, which is most accurate once estimates concentrate and the surrogate decision has controlled curvature; early in learning, variance estimates may be unstable, and the choice of surrogate/temperature introduces a bias–smoothness trade-off. In RL and planning, trajectory coupling and bootstrapping create dependencies that are only approximately captured by the diagonal variance view, suggesting room for sharper analysis. The RL guarantee in this paper is asymptotic rather than a finite-sample fixed-budget BPI bound.

**Future work.** Promising directions include:

- decision-variance control with correlated uncertainty (full covariance);
- adaptive temperature schedules that automatically balance bias and variance;
- higher-order allocation rules using Hessian and moment information when the additional estimation cost is justified;
- non-asymptotic theory for variance-driven RL;
- scalable influence estimation under function approximation.

## Impact Statement

This paper presents work whose goal is to advance the field of Machine Learning. There are many potential societal consequences of our work, none which we feel must be specifically highlighted here.

## Acknowledgements

This research is funded by Vietnam National Foundation for Science and Technology Development (NAFOSTED) under grant number 102.01-2025.47. The author Hung The Tran is funded by the Quantum AI & Cyber Security Institute, FPT Corporation.

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

# Appendix

This appendix provides additional proofs, full algorithm pseudo-codes, and experimental details referenced in the main text. Unless stated otherwise, notation matches the main text.

## Contents

## A. Proofs for Section 3 (VarDE Methodology)

### A.1. Proof of Lemma 3.2 (Local Linear Approximation)

*Proof.* Let $\Delta := \hat{\mu} - \mu$. By Taylor's theorem with remainder on the segment between $\mu$ and $\hat{\mu}$, which lies in $D = [a, b]^n$ under Assumption 3.1,

$$f(\hat{\mu}) = f(\mu) + \nabla f(\mu)^\top \Delta + \frac{1}{2}\Delta^\top H_f(\xi)\Delta,$$

for some $\xi$ on the segment $[\mu, \hat{\mu}]$. Since $\|H_f(\xi)\|_{\mathrm{op}} \leq M$,

$$\left|\frac{1}{2}\Delta^\top H_f(\xi)\Delta\right| \leq \frac{1}{2}\|H_f(\xi)\|_{\mathrm{op}}\|\Delta\|_2^2 \leq \frac{M}{2}\|\Delta\|_2^2.$$

Thus the remainder is $\mathcal{O}(M\|\hat{\mu} - \mu\|_2^2)$. Writing $w_i(\mu) = \partial_i f(\mu)$ and $c = f(\mu) - \nabla f(\mu)^\top \mu$ gives the stated first-order form. $\square$

### A.2. Auxiliary min-pulls guarantee

Throughout the proof, write

$$\Delta(t) := \hat{\mu}(t) - \mu, \qquad \Delta_i(t) := \hat{\mu}_i(t) - \mu_i, \qquad Y_t := f(\hat{\mu}(t)),$$

and let $g := \nabla f(\mu)$, so that $g_i = w_i(\mu)$. Under Assumption 3.1, bounded observations imply $\mu, \hat{\mu}(t) \in D = [a, b]^n$ for all $t$, and the segment between them is contained in $D$. Moreover, because each $w_i$ is continuous on the compact set $D$,

$$W := \max_{x \in D} \max_{i \in [n]} |w_i(x)| < \infty.$$

**Lemma A.1** (General min-pulls guarantee). *Under Assumption 3.1, the variance floor in Section 3.4, and Assumption 3.3, there exist constants $c > 0$ and $C > 0$, depending only on $n, \rho, W, \bar{\sigma}^2, a, b$, such that for all $t \geq n$ and all $i \in [n]$,*

$$N_i(t) \geq c\frac{t}{n} - C.$$

*In particular, $N_i(t) = \Omega(t)$ for every component.*

*Proof.* Let $\mathcal{M}$ be a component with maximal pull count at time $t$, so $N_{\mathcal{M}}(t) \geq t/n$. Let $t' \leq t$ be the last time at which $\mathcal{M}$ is selected. Immediately before that pull, the VarDE rule implies that for every $i \neq \mathcal{M}$,

$$\frac{w_{\mathcal{M}}(\hat{\mu}(t' - 1))^2 \tilde{\sigma}_{\mathcal{M}}^2(t' - 1)}{N_{\mathcal{M}}(t' - 1)(N_{\mathcal{M}}(t' - 1) + 1)} \geq \frac{w_i(\hat{\mu}(t' - 1))^2 \tilde{\sigma}_i^2(t' - 1)}{N_i(t' - 1)(N_i(t' - 1) + 1)}.$$

Rearranging gives

$$\frac{N_i(t'-1)(N_i(t'-1)+1)}{N_{\mathcal{M}}(t'-1)(N_{\mathcal{M}}(t'-1)+1)} \geq \frac{w_i(\hat{\mu}(t'-1))^2}{w_{\mathcal{M}}(\hat{\mu}(t'-1))^2} \cdot \frac{\tilde{\sigma}_i^2(t'-1)}{\tilde{\sigma}_{\mathcal{M}}^2(t'-1)}.$$

By Assumption 3.3 and by the definition of $W$,

$$\frac{w_i(\hat{\mu}(t'-1))^2}{w_{\mathcal{M}}(\hat{\mu}(t'-1))^2} \geq \frac{\rho^2}{W^2}.$$

Also, the variance floor and bounded observations give

$$\tilde{\sigma}_i^2(t'-1) \geq \bar{\sigma}^2, \qquad \tilde{\sigma}_{\mathcal{M}}^2(t'-1) \leq \frac{(b-a)^2}{4}.$$

Hence

$$\frac{N_i(t'-1)(N_i(t'-1)+1)}{N_{\mathcal{M}}(t'-1)(N_{\mathcal{M}}(t'-1)+1)} \geq \frac{\rho^2}{W^2} \cdot \frac{4\bar{\sigma}^2}{(b-a)^2} =: c_0 > 0.$$

Since $x \mapsto x(x+1)$ is increasing on $[0, \infty)$, there exist constants $c_1, C_1 > 0$ depending only on $c_0$ such that

$$x(x+1) \geq c_0 y(y+1) \implies x \geq c_1 y - C_1.$$

Therefore

$$N_i(t'-1) \geq c_1 N_{\mathcal{M}}(t'-1) - C_1.$$

Counts are nondecreasing and $t'$ is the last time at which $\mathcal{M}$ is selected, so

$$N_i(t) \geq N_i(t'-1), \qquad N_{\mathcal{M}}(t'-1) = N_{\mathcal{M}}(t) - 1 \geq \frac{t}{n} - 1.$$

Combining these inequalities yields

$$N_i(t) \geq c_1 \left(\frac{t}{n} - 1\right) - C_1 = c\frac{t}{n} - C$$

for suitable constants $c, C > 0$. $\qquad \square$

**Corollary A.2.** *Under the assumptions of Lemma A.1, for every fixed integer $k \in \{1, 2, 3\}$,*

$$\sum_{i=1}^{n} N_i(t)^{-k} = O(t^{-k}), \qquad \sum_{i=1}^{n} \frac{\sigma_i^2}{N_i(t)} = O(t^{-1}).$$

*Proof.* Lemma A.1 gives $N_i(t) \geq ct/n - C$ for all $i$. Since $n$ is fixed, each $N_i(t)^{-k} = O(t^{-k})$, and summing over $i$ proves the first claim. The second follows immediately because the variances are finite under bounded observations. $\qquad \square$

### A.3. Proof of Lemma 3.4 (First-Order Variance Decomposition)

*Proof.* We prove the stated expansion along a VarDE trajectory under Assumption 3.1, the variance floor in Section 3.4, and Assumption 3.3.

Fix $t$ and write $\Delta := \Delta(t)$, $\hat{\mu} := \hat{\mu}(t)$, and $Y := Y_t$. By Taylor's theorem,

$$Y = f(\mu) + g^\top \Delta + Q, \qquad Q := \frac{1}{2}\Delta^\top H_f(\xi)\Delta,$$

for some random point $\xi$ on the segment $[\mu, \hat{\mu}]$. Assumption 3.1 applies to the whole segment. Therefore

$$\text{Var}(Y) = \text{Var}(g^\top \Delta) + \text{Var}(Q) + 2\,\text{Cov}(g^\top \Delta, Q).$$

**Leading term.** Since the components are independent,

$$\text{Var}(g^\top \Delta) = \sum_{i=1}^{n} g_i^2 \frac{\sigma_i^2}{N_i(t)} = \sum_{i=1}^{n} w_i(\mu)^2 \frac{\sigma_i^2}{N_i(t)}.$$

**Quadratic remainder.** By the bounded Hessian condition,

$$|Q| \le \frac{M}{2} \|\Delta\|_2^2,$$

where $M := \sup_{z \in [a,b]^n} \|H_f(z)\|_{\mathrm{op}}$. Hence

$$\mathrm{Var}(Q) \le \mathbb{E}[Q^2] \le \frac{M^2}{4} \mathbb{E} \|\Delta\|_2^4.$$

Using $(\sum_i a_i)^2 \le n \sum_i a_i^2$ with $a_i = \Delta_i^2$,

$$\|\Delta\|_2^4 \le n \sum_{i=1}^n \Delta_i^4.$$

For bounded observations, there are constants $C_{4,i} < \infty$ such that $\mathbb{E}[\Delta_i^4] \le C_{4,i} N_i(t)^{-2}$. By Corollary A.2,

$$\mathbb{E} \|\Delta\|_2^4 = O(t^{-2}), \qquad \mathrm{Var}(Q) = O(t^{-2}).$$

**Covariance term.** Write

$$Q_0 := \frac{1}{2} \Delta^\top H_f(\mu) \Delta, \qquad Q_1 := Q - Q_0.$$

Then

$$\mathrm{Cov}(g^\top \Delta, Q) = \mathrm{Cov}(g^\top \Delta, Q_0) + \mathrm{Cov}(g^\top \Delta, Q_1).$$

For $Q_0$, since $H_f(\mu)$ is deterministic and $\mathbb{E}[g^\top \Delta] = 0$,

$$\mathrm{Cov}(g^\top \Delta, Q_0) = \mathbb{E}[g^\top \Delta \, Q_0] = \frac{1}{2} \sum_{i,j,k} g_k (H_f(\mu))_{ij} \mathbb{E}[\Delta_k \Delta_i \Delta_j].$$

By independence and centering, every mixed term vanishes unless $i = j = k$. Hence

$$\mathrm{Cov}(g^\top \Delta, Q_0) = \frac{1}{2} \sum_{i=1}^n g_i (H_f(\mu))_{ii} \mathbb{E}[\Delta_i^3].$$

Bounded observations imply $|\mathbb{E}[\Delta_i^3]| \le C_{3,i} N_i(t)^{-2}$, and therefore

$$|\mathrm{Cov}(g^\top \Delta, Q_0)| = O(t^{-2}).$$

For $Q_1$, Lipschitz continuity of the Hessian gives

$$\|H_f(\xi) - H_f(\mu)\|_{\mathrm{op}} \le L\|\xi - \mu\|_2 \le L\|\Delta\|_2,$$

so

$$|Q_1| \le \frac{L}{2} \|\Delta\|_2^3.$$

By Cauchy–Schwarz,

$$|\mathrm{Cov}(g^\top \Delta, Q_1)| \le \sqrt{\mathrm{Var}(g^\top \Delta) \, \mathbb{E}[Q_1^2]}.$$

Now $\mathrm{Var}(g^\top \Delta) = O(t^{-1})$ by Corollary A.2. Also,

$$\mathbb{E}[Q_1^2] \le \frac{L^2}{4} \mathbb{E} \|\Delta\|_2^6.$$

Using $(\sum_i a_i)^3 \le n^2 \sum_i a_i^3$ with $a_i = \Delta_i^2$ and the bounded-observation moment bound $\mathbb{E}|\Delta_i|^6 \le C_{6,i} N_i(t)^{-3}$,

$$\mathbb{E} \|\Delta\|_2^6 = O(t^{-3}).$$

Thus

$$|\mathrm{Cov}(g^\top \Delta, Q_1)| = O(t^{-2}).$$

Combining the leading term, $\mathrm{Var}(Q) = O(t^{-2})$, and both covariance bounds gives

$$\mathrm{Var}(Y_t) = \sum_{i=1}^n w_i(\mu)^2 \frac{\sigma_i^2}{N_i(t)} + O(t^{-2}).$$

$\square$

## A.4. Proof of Lemma 3.5 (One-Sample Variance Decrement)

*Proof.* Let $\hat{\mu}_i^{(N)} = N^{-1} \sum_{s=1}^{N} X_{i,s}$ with $\mathrm{Var}[X_{i,s}] = \sigma_i^2$. Then

$$\mathrm{Var}[\hat{\mu}_i^{(N)}] = \frac{\sigma_i^2}{N}, \qquad \mathrm{Var}[\hat{\mu}_i^{(N+1)}] = \frac{\sigma_i^2}{N+1}.$$

Therefore

$$\Delta\,\mathrm{Var}[\hat{\mu}_i] = \mathrm{Var}[\hat{\mu}_i^{(N+1)}] - \mathrm{Var}[\hat{\mu}_i^{(N)}] = -\frac{\sigma_i^2}{N(N+1)}.$$

$\square$

## A.5. Proof of Proposition 3.6 (Influence-Weighted Global Variance Decrement)

*Proof.* By Lemma 3.4,

$$V_t := \mathrm{Var}[Y_t] = \sum_{j=1}^{n} w_j(\mu)^2 \frac{\sigma_j^2}{N_j(t)} + O(t^{-2}),$$

and similarly

$$V_{t+1} = \sum_{j=1}^{n} w_j(\mu)^2 \frac{\sigma_j^2}{N_j(t+1)} + O((t+1)^{-2}).$$

If component $i_t$ is sampled, then $N_{i_t}(t+1) = N_{i_t}(t) + 1$ and all other counts are unchanged. Hence

$$\sum_{j=1}^{n} w_j(\mu)^2 \left( \frac{\sigma_j^2}{N_j(t+1)} - \frac{\sigma_j^2}{N_j(t)} \right) = -\frac{w_{i_t}(\mu)^2 \sigma_{i_t}^2}{N_{i_t}(t)(N_{i_t}(t)+1)}.$$

Since $O((t+1)^{-2}) - O(t^{-2}) = O(t^{-2})$, subtracting the two expansions gives

$$V_{t+1} - V_t = -\frac{w_{i_t}(\mu)^2 \sigma_{i_t}^2}{N_{i_t}(t)(N_{i_t}(t)+1)} + O(t^{-2}).$$

$\square$

## A.6. Proof of Theorem 3.7 (Variance Decay of VarDE)

*Proof.* Lemma A.1 gives $N_i(T) = \Omega(T)$ for every $i \in [n]$. By Lemma 3.4,

$$\mathrm{Var}[Y_T] = \sum_{i=1}^{n} w_i(\mu)^2 \frac{\sigma_i^2}{N_i(T)} + O(T^{-2}).$$

The leading term is $O(T^{-1})$ because every $N_i(T)$ is linear in $T$, while the curvature correction is $O(T^{-2})$. Thus

$$\mathrm{Var}[Y_T] = O(T^{-1}) + O(T^{-2}) = O(T^{-1}).$$

$\square$

## A.7. First-order allocation constant

The same decomposition also clarifies what VarDE can improve asymptotically. Suppose an allocation has limiting proportions

$$N_i(T) = p_i T + r_i(T), \qquad r_i(T) = O(1), \qquad p_i > 0, \qquad \sum_{i=1}^{n} p_i = 1.$$

For each fixed $i$,

$$\frac{1}{N_i(T)} = \frac{1}{p_i T + r_i(T)} = \frac{1}{p_i T} + O(T^{-2}),$$

so Lemma 3.4 gives the sharp first-order expansion

$$\mathrm{Var}[Y_T] = \sum_{i=1}^{n} w_i(\mu)^2 \frac{\sigma_i^2}{N_i(T)} + O(T^{-2}) = \frac{1}{T}C(p) + O(T^{-2}),$$

where

$$C(p) := \sum_{i=1}^{n} \frac{w_i(\mu)^2 \sigma_i^2}{p_i}.$$

Thus every allocation with nonzero limiting proportions has the same $T^{-1}$ exponent; the meaningful asymptotic comparison is the leading constant $C(p)$.

To minimize $C(p)$ over the probability simplex, apply Cauchy–Schwarz:

$$\left(\sum_i |w_i(\mu)|\sigma_i\right)^2 = \left(\sum_i \frac{|w_i(\mu)|\sigma_i}{\sqrt{p_i}}\sqrt{p_i}\right)^2 \leq \left(\sum_i \frac{w_i(\mu)^2 \sigma_i^2}{p_i}\right)\left(\sum_i p_i\right) = C(p).$$

Equality holds if and only if

$$\frac{|w_i(\mu)|\sigma_i}{\sqrt{p_i}} = c\sqrt{p_i} \quad \text{for all } i,$$

for some constant $c > 0$, equivalently $p_i \propto |w_i(\mu)|\sigma_i$. Hence the optimal first-order allocation and constant are

$$p_i^\star = \frac{|w_i(\mu)|\sigma_i}{\sum_j |w_j(\mu)|\sigma_j}, \qquad C_\star = \left(\sum_i |w_i(\mu)|\sigma_i\right)^2.$$

For uniform allocation $p_i = 1/n$, the constant is

$$C_{\mathrm{unif}} = n\sum_i w_i(\mu)^2 \sigma_i^2,$$

and $C_\star \leq C_{\mathrm{unif}}$, with equality only in the degenerate case where $|w_i(\mu)|\sigma_i$ is constant across $i$.

This is the sense in which VarDE is designed to improve the asymptotics. Its empirical one-step score is

$$S_i(t) = w_i(\hat{\mu}(t))^2 \frac{\tilde{\sigma}_i^2(t)}{N_i(t)(N_i(t)+1)}.$$

In the stabilized regime, $w_i(\hat{\mu}(t)) \approx w_i(\mu)$, $\tilde{\sigma}_i^2(t) \approx \sigma_i^2$, and $N_i(t)(N_i(t)+1) \approx N_i(t)^2$, so

$$S_i(t) \approx \frac{w_i(\mu)^2 \sigma_i^2}{N_i(t)^2}.$$

A greedy allocation that repeatedly samples the largest marginal decrement tends to equalize these leading marginal scores among persistently sampled components. Equalization gives

$$\frac{w_i(\mu)^2 \sigma_i^2}{N_i(T)^2} \approx \frac{w_j(\mu)^2 \sigma_j^2}{N_j(T)^2},$$

and therefore

$$\frac{N_i(T)}{N_j(T)} \approx \frac{|w_i(\mu)|\sigma_i}{|w_j(\mu)|\sigma_j}.$$

After normalization by $T$, this is exactly the optimal first-order proportion $p^\star$. Hence VarDE's intended advantage is not a better variance-decay exponent than all well-spread allocations, but a smaller first-order constant by targeting the allocation that minimizes the decision-level variance expansion.

# B. Proofs for Subsection 4.1 (VARDE–BAI)

### B.1. Proof of Lemma 4.1 (LSE bound)

*Proof.* *Lower bound.* Since $\sum_{i=1}^{K} e^{\hat{\mu}_i/\tau} \geq e^{\max_i \hat{\mu}_i/\tau}$,

$$\mathrm{LSE}_\tau(\hat{\mu}) = \tau \log \sum_{i=1}^{K} e^{\hat{\mu}_i/\tau} \;\geq\; \tau \log e^{\max_i \hat{\mu}_i/\tau} \;=\; \max_i \hat{\mu}_i.$$

*Upper bound.* Also $\sum_{i=1}^{K} e^{\hat{\mu}_i/\tau} \leq K \cdot e^{\max_i \hat{\mu}_i/\tau}$, hence

$$\mathrm{LSE}_\tau(\hat{\mu}) = \tau \log \sum_{i=1}^{K} e^{\hat{\mu}_i/\tau} \;\leq\; \tau \log\!\Big( K \, e^{\max_i \hat{\mu}_i/\tau} \Big) \;=\; \max_i \hat{\mu}_i + \tau \log K.$$

This proves the claim. $\qquad\square$

### B.2. Proof of Lemma 4.2 (LSE curvature)

*Proof.* Let $f(x) = \mathrm{LSE}_\tau(x) = \tau \log \sum_{i=1}^{K} e^{x_i/\tau}$. Then

$$\frac{\partial f}{\partial x_i}(x) = \frac{e^{x_i/\tau}}{\sum_{j=1}^{K} e^{x_j/\tau}} = p_i(x), \qquad i = 1, \dots, K,$$

Differentiating once more,

$$\nabla^2 f(x) = \frac{1}{\tau}\big(\mathrm{Diag}(p(x)) - p(x)p(x)^\top\big).$$

For any $v \in \mathbb{R}^K$ satisfying $\|v\|_2 = 1$, we have

$$\begin{aligned}
v^\top \nabla^2 f(x)\, v &= \frac{1}{\tau}\left( \sum_{i=1}^{K} p_i(x)\, v_i^2 - \Big( \sum_{i=1}^{K} p_i(x)\, v_i \Big)^2 \right) \\
&= \frac{1}{\tau}\, \mathrm{Var}_{i \sim p(x)}[v_i] \\
&\leq \frac{1}{\tau}\, \frac{(\max_i v_i - \min_i v_i)^2}{4} \\
&\leq \frac{1}{\tau}\, \frac{(\max_i v_i)^2 + (\min_i v_i)^2}{2} \\
&\leq \frac{1}{\tau}\, \frac{\sum_{i=1}^{K} v_i^2}{2} \\
&= \frac{1}{2\tau}.
\end{aligned}$$

Taking the supremum over all $\|v\|_2 = 1$ gives

$$\|\nabla^2 f(x)\|_{\mathrm{op}} = \sup_{\|v\|_2=1} v^\top \nabla^2 f(x)\, v \leq \frac{1}{2\tau} \quad \forall x.$$

$\qquad\square$

### B.3. Proof of Lemma 4.3 (LSE Hessian Lipschitzness)

*Proof.* Let

$$p_i(x) = \frac{e^{x_i/\tau}}{\sum_{\ell=1}^{K} e^{x_\ell/\tau}}.$$

As in the proof of Lemma 4.2, the Hessian entries are

$$H_{ij}(x) = \frac{1}{\tau} \left( \delta_{ij} p_i(x) - p_i(x) p_j(x) \right).$$

The softmax derivative is

$$\partial_k p_i(x) = \frac{1}{\tau} p_i(x)(\delta_{ik} - p_k(x)).$$

Therefore

$$\partial_k H_{ij}(x) = \frac{1}{\tau^2} \left[ \delta_{ij} p_i(\delta_{ik} - p_k) - p_i(\delta_{ik} - p_k)p_j - p_i p_j(\delta_{jk} - p_k) \right].$$

Since $p_i \in [0, 1]$ for all $i$, each bracketed term has absolute value at most one, and hence

$$|\partial_k H_{ij}(x)| \leq \frac{3}{\tau^2}$$

uniformly over $x \in \mathbb{R}^K$ and all $i, j, k$. For any $x, y \in D$, the scalar mean-value theorem applied entrywise gives

$$|H_{ij}(x) - H_{ij}(y)| \leq \frac{3}{\tau^2} \sum_{k=1}^{K} |x_k - y_k| \leq \frac{3\sqrt{K}}{\tau^2} \|x - y\|_2.$$

Thus

$$
\begin{aligned}
\|H(x) - H(y)\|_{\mathrm{op}} &\leq \|H(x) - H(y)\|_{\mathrm{F}} \\
&\leq K \max_{i,j} |H_{ij}(x) - H_{ij}(y)| \\
&\leq \frac{3K^{3/2}}{\tau^2} \|x - y\|_2.
\end{aligned}
$$

This proves the claim with $L_{\tau,K} = 3K^{3/2}/\tau^2$. $\qquad\square$

## B.4. Proof of Lemma 4.4 (LSE nonvanishing influence weights)

*Proof.* For any $x \in D = [a, b]^K$, each coordinate satisfies $x_i \geq a$, so $e^{x_i/\tau} \geq e^{a/\tau}$. Likewise, $x_j \leq b$ for every $j$, so

$$\sum_{j=1}^{K} e^{x_j/\tau} \leq K e^{b/\tau}.$$

Therefore

$$w_i(x) = \frac{e^{x_i/\tau}}{\sum_{j=1}^{K} e^{x_j/\tau}} \geq \frac{e^{a/\tau}}{K e^{b/\tau}} = \frac{1}{K} e^{(a-b)/\tau} = \rho_{\tau,D} > 0.$$

The upper bound $w_i(x) \leq 1$ is immediate from the definition of the softmax weights. This proves the claim. $\qquad\square$

## B.5. Proof of Theorem 4.5 (VARDE–BAI error probability)

**Lemma B.1** (Confidence interval for the empirical variance). *Let $X_1, \ldots, X_n$ be i.i.d. random variables supported on $[a, b]$, with $\mu = \mathbb{E}[X]$ and $\sigma^2 = \mathrm{Var}(X)$. Define*

$$\hat{\mu} = \frac{1}{n} \sum_{i=1}^{n} X_i, \qquad \hat{\sigma}^2 = \frac{1}{n} \sum_{i=1}^{n} (X_i - \hat{\mu})^2.$$

*Then for any $\delta \in (0, 1)$, with probability at least $1 - \delta$,*

$$\left| \hat{\sigma}^2 - \sigma^2 \right| \leq (b-a)^2 \left( \sqrt{\frac{\log(2/\delta)}{n}} + \frac{\log(2/\delta)}{n} \right).$$

*Equivalently, for any $k > 0$, with probability at least $1 - 2e^{-k}$,*

$$\left| \hat{\sigma}^2 - \sigma^2 \right| \leq (b-a)^2 \left( \sqrt{\frac{k}{n}} + \frac{k}{n} \right).$$

*Proof.* We expand the empirical variance as

$$\hat{\sigma}^2 = \frac{1}{n} \sum_{i=1}^{n} (X_i - \hat{\mu})^2$$

$$= \frac{1}{n} \sum_{i=1}^{n} \Big[ (X_i - \mu) - (\hat{\mu} - \mu) \Big]^2$$

$$= \frac{1}{n} \sum_{i=1}^{n} \Big[ (X_i - \mu)^2 - 2(X_i - \mu)(\hat{\mu} - \mu) + (\hat{\mu} - \mu)^2 \Big]$$

$$= \frac{1}{n} \sum_{i=1}^{n} (X_i - \mu)^2 - 2(\hat{\mu} - \mu) \frac{1}{n} \sum_{i=1}^{n} (X_i - \mu) + (\hat{\mu} - \mu)^2.$$

$$= \frac{1}{n} \sum_{i=1}^{n} (X_i - \mu)^2 - 2(\hat{\mu} - \mu)^2 + (\hat{\mu} - \mu)^2$$

$$= \frac{1}{n} \sum_{i=1}^{n} (X_i - \mu)^2 - (\hat{\mu} - \mu)^2.$$

Thus, we can write the error as

$$\hat{\sigma}^2 - \sigma^2 = \left( \frac{1}{n} \sum_{i=1}^{n} (X_i - \mu)^2 - \sigma^2 \right) - (\hat{\mu} - \mu)^2$$

$$= \underbrace{\frac{1}{n} \sum_{i=1}^{n} \Big[ (X_i - \mu)^2 - \sigma^2 \Big]}_{A} - \underbrace{(\hat{\mu} - \mu)^2}_{B}.$$

Since $X_i \in [a, b]$, it follows that $(X_i - \mu)^2 \in [0, (b-a)^2]$. Applying Hoeffding's inequality to $(X_i - \mu)^2$ gives

$$\Pr \left( |A| \le (b-a)^2 \sqrt{\frac{\log(2/\delta)}{n}} \right) \ge 1 - \frac{\delta}{2}.$$

Next apply Hoeffding to $\hat{\mu}$:

$$\Pr \left( |\hat{\mu} - \mu| \le (b-a) \sqrt{\frac{\log(2/\delta)}{n}} \right) \ge 1 - \frac{\delta}{2}.$$

Hence

$$B = (\hat{\mu} - \mu)^2 \le (b-a)^2 \frac{\log(2/\delta)}{n}.$$

By a union bound, with probability at least $1 - \delta$, both bounds on $A$ and $B$ hold. Therefore,

$$\left| \hat{\sigma}^2 - \sigma^2 \right| \le |A| + |B| \le (b-a)^2 \left( \sqrt{\frac{\log(2/\delta)}{n}} + \frac{\log(2/\delta)}{n} \right).$$

$\square$

**Lemma B.2** (Lower bound on each arm's number of pulls)**.** *Assume rewards are supported on* $[0, 1]$. *Run* VARDE−BAI *(Alg. 1) with temperature* $\tau > 0$ *and warm start* $\eta$. *Then, after* $T > K$ *total pulls, each arm* $i$ *has been pulled at least*

$$N_i(T) \ge 2e^{-1/\tau} \bar{\sigma} \left( \frac{T}{K} - \frac{1}{2} \right) - \frac{1}{2}.$$

*Proof.* At each round $t$, we choose the arm $i$ with the maximum value

$$\arg\max_i \frac{w_i(\hat{\mu}(t))^2 \, \tilde{\sigma}_i^2(t)}{N_i(t)\big(N_i(t)+1\big)},$$

where

$$w_i(\hat{\mu}(t)) = \frac{\exp\big(\hat{\mu}_i(t)/\tau\big)}{\sum_j \exp\big(\hat{\mu}_j(t)/\tau\big)}, \qquad \tilde{\sigma}_i(t) := \max\big\{\hat{\sigma}_i(t), \bar{\sigma}\big\}.$$

Assume that after all $T$ rounds, the arm $M$ has been chosen with maximum times. At the final time $t$ where arm $M$ is selected, the selection rule implies that

$$\frac{w_M(\hat{\mu}(t))^2 \, \tilde{\sigma}_M^2(t)}{N_M(t)\big(N_M(t)+1\big)} \;\geq\; \frac{w_i(\hat{\mu}(t))^2 \, \tilde{\sigma}_i^2(t)}{N_i(t)\big(N_i(t)+1\big)}, \qquad \forall i \neq M.$$

Rearranging the above inequality yields

$$\frac{N_i(t)\big(N_i(t)+1\big)}{N_M(t)\big(N_M(t)+1\big)} \;\geq\; \exp\!\Big(\frac{2}{\tau}\big(\hat{\mu}_i(t)-\hat{\mu}_M(t)\big)\Big) \cdot \frac{\tilde{\sigma}_i^2(t)}{\tilde{\sigma}_M^2(t)}.$$

Since rewards lie in $[0,1]$, empirical means satisfy $\hat{\mu}_k(t) \in [0,1]$. Consequently, $\hat{\mu}_i(t)-\hat{\mu}_M(t) \geq -1, \forall\, i \neq M$. Thus,

$$\frac{N_i(t)\big(N_i(t)+1\big)}{N_M(t)\big(N_M(t)+1\big)} \;\geq\; e^{-2/\tau} \cdot \frac{\tilde{\sigma}_i^2(t)}{\tilde{\sigma}_M^2(t)}.$$

Since we have $1/4 \geq \tilde{\sigma}_i^2(t) \geq \bar{\sigma}^2$, it follows that $\tilde{\sigma}_i^2(t)/\tilde{\sigma}_M^2(t) \geq 4\bar{\sigma}^2$. Therefore,

$$\frac{N_i(t)\big(N_i(t)+1\big)}{N_M(t)\big(N_M(t)+1\big)} \;\geq\; 4e^{-2/\tau} \cdot \bar{\sigma}^2.$$

Since $N_i(t), N_M(t) \geq 0$, we also have

$$\frac{N_i^2(t)+N_i(t)+1/4}{N_M^2(t)+N_M(t)+1/4} \;\geq\; \frac{N_i^2(t)+N_i(t)}{N_M^2(t)+N_M(t)} \;\geq\; 4e^{-2/\tau} \cdot \bar{\sigma}^2.$$

Which is equivalent to

$$\left(\frac{N_i(t)+1/2}{N_M(t)+1/2}\right)^2 \;\geq\; 4e^{-2/\tau} \cdot \bar{\sigma}^2.$$

So that

$$N_i(t) \;\geq\; 2e^{-1/\tau}\bar{\sigma}\left(N_M(t)+\frac{1}{2}\right) - \frac{1}{2}.$$

Since $M$ is pulled most frequently after $T$ rounds and $t$ is the last time it was pulled, $N_M(t) = N_M(T) - 1 \geq T/K - 1$. Thus,

$$N_i(t) \;\geq\; 2e^{-1/\tau}\bar{\sigma}\left(\frac{T}{K}-\frac{1}{2}\right) - \frac{1}{2}.$$

This inequality holds for all arms $i$, including $M$, which is the trivial case. $\qquad\square$

***Proof of Theorem 4.5.*** We write the empirical gap between the best arm $i^*$ and arm $i \neq i^*$ after $T$ total pulls as

$$\hat{\mu}_{i^*}(T) - \hat{\mu}_i(T) = \frac{1}{N_{i^*}(T)}\sum_{n=1}^{N_{i^*}(T)} X_{i^*,n} - \frac{1}{N_i(T)}\sum_{n=1}^{N_i(T)} X_{i,n}.$$

Since $\mathbb{E}[\hat{\mu}_{i^*}(T) - \hat{\mu}_i(T)] = \mu_{i^*} - \mu_i = \Delta_i$ and the samples are independent, we can apply Hoeffding's inequality to get

$$\Pr\{\hat{\mu}_i(T) - \hat{\mu}_{i^*}(T) + \Delta_i \geq \Delta_i\} \leq \exp\!\left(-\frac{2\Delta_i^2}{\dfrac{1}{N_i(T)} + \dfrac{1}{N_{i^*}(T)}}\right).$$

Using Lemma B.2 and rearranging the terms gives

$$\Pr\{\hat{\mu}_i(T) \geq \hat{\mu}_{i^*}(T)\} \leq \exp\left[-\Delta_i^2 \left(2e^{-1/\tau}\bar{\sigma}\left(\frac{T}{K} - \frac{1}{2}\right) - \frac{1}{2}\right)\right].$$

The error probability of VARDE–BAI is obtained by a union bound over all suboptimal arms:

$$\Pr\{\hat{i}_T \neq i^*\} \leq \sum_{i \neq i^*} \Pr\{\hat{\mu}_i(T) \geq \hat{\mu}_{i^*}(T)\} \leq \sum_{i \neq i^*} \exp\left[-\Delta_i^2 \left(2e^{-1/\tau}\bar{\sigma}\left(\frac{T}{K} - \frac{1}{2}\right) - \frac{1}{2}\right)\right].$$

$\square$

## C. Proofs for Subsection 4.2 (VARDE–MCTS)

Throughout this section, rewards are in $[0, 1]$, the horizon is $H$, hence returns are therefore bounded in $[0, H]$. For each node $s$ and action $a \in \mathcal{A}(s)$, we denote that $Q^*(s, a)$ is the true action-value, $\hat{Q}(s, a)$ is its empirical estimate by back-propagation, $N(s)$ is the number of visits to node $s$, $N(s, a)$ is the number of times action $a$ has been selected at node $s$, and $\mathcal{S}(s, a) = \{s' : P(s' \mid s, a) > 0\}$ is the (finite) set of possible successor states.

### C.1. Reduction to VARDE–BAI at each node

Recall the selection rule of VARDE–MCTS (Algorithm 2): at a node $s_h$ with empirical estimates $\hat{Q}(s_h, \cdot)$, empirical variances $\hat{\sigma}^2(s_h, \cdot)$ and counts $N(s_h, \cdot)$, we first compute influence weights

$$w_\tau(s_h, a) := \frac{\exp(\hat{Q}(s_h, a)/\tau)}{\sum_{b \in A(s_h)} \exp(\hat{Q}(s_h, b)/\tau)},$$

and then select

$$a_h \in \arg \max_{a \in A(s_h)} \frac{w_\tau(s_h, a)^2 \, \tilde{\sigma}^2(s_h, a)}{N(s_h, a) \left(N(s_h, a) + 1\right)}.$$

This is exactly the VARDE–BAI selection rule applied to a bandit whose arms correspond to actions $a \in A(s_h)$, with empirical means $\hat{\mu}_i \equiv \hat{Q}(s_h, a)$, empirical variances $\hat{\sigma}_i^2 \equiv \hat{\sigma}^2(s_h, a)$ and counts $N_i \equiv N(s_h, a)$.

Therefore, we also have the following lower bound on the number of times each action selected at node $s$:

**Lemma C.1** (Min-pulls per action at each node). *Run VARDE–MCTS (Alg. 2) with temperature $\tau$. Then, after $N(s) > |\mathcal{A}(s)|$ visits to node $s$, each action $a \in \mathcal{A}(s)$ has been selected at least*

$$N(s, a) \geq 2e^{-1/\tau} \bar{\sigma} \left(\frac{N(s)}{|\mathcal{A}(s)|} - \frac{1}{2}\right) - \frac{1}{2}.$$

**Corollary C.2.** *For any state $s$ with $N(s) > |\mathcal{A}(s)|$ visits and any action $a \in \mathcal{A}(s)$, exist an $\alpha > 0$ such that*

$$N(s, a) \geq \alpha N(s)$$

### C.2. Concentration of empirical reward mean

**Lemma C.3** (Concentration of empirical reward mean). *Fix any state $s$ and action $a \in \mathcal{A}(s)$ with $N(s, a)$ visits, the $i^{th}$ visit receives reward $\hat{r}_i(s, a)$. Let $\bar{r}(s, a) = \frac{1}{N(s,a)} \sum_{i=1}^{N(s,a)} \hat{r}_i(s, a)$ be the empirical mean of the rewards, and $r(s, a)$ the true expected reward. Then, for any $\varepsilon > 0$, conditional on $N(s, a)$ we have*

$$\Pr \left(|\bar{r}(s, a) - r(s, a)| > \varepsilon\right) \leq 2 \exp\left(-2\,\varepsilon^2 N(s, a)\right)$$

*Proof.* Since the rewards are independently sampled from identical distribution $\nu(\cdot|s, a)$, this lemma can be obtained using Hoeffding's inequality. $\square$

### C.3. Concentration of empirical transitions

**Lemma C.4** (Concentration of empirical transitions). *Fix any state $s$ and action $a \in \mathcal{A}(s)$. For each $s' \in \mathcal{S}(s, a)$ let $N(s')$ be the number of observed transitions from $(s, a)$ to $s'$ and recall that $\hat{P}(s' \mid s, a) := N(s')/N(s, a)$. Then, for any $\varepsilon > 0$, conditional on $N(s, a)$ we have*

$$\Pr \left(\max_{s' \in \mathcal{S}(s,a)} |\hat{P}(s' \mid s, a) - P(s' \mid s, a)| > \varepsilon\right) \leq 2 \exp\left(-\frac{1}{2}\,\varepsilon^2 N(s, a)\right).$$

*Proof.* Fix $(s, a)$ and condition on $N(s, a) = n > 0$. Each time VARDE–MCTS visits $(s, a)$, the next state is drawn from the true kernel $P(\cdot \mid s, a)$. The visit times are chosen adaptively by the tree policy, but given $n$ the $n$ successor states of $(s, a)$ are i.i.d. with distribution $P(\cdot \mid s, a)$. Hence we may work with an i.i.d. sample of size $n$.

Enumerate the (finite) successor set as
$$\mathcal{S}(s,a) = \{s_1, \ldots, s_m\}.$$
For $j = 1, \ldots, n$, let $Y_j \in \{1, \ldots, m\}$ be the index of the $j$-th observed successor of $(s,a)$:
$$Y_j = k \quad \Longleftrightarrow \quad \text{the } j\text{-th successor is } s_k.$$

Then $Y_1, \ldots, Y_n$ are i.i.d. and
$$\Pr(Y_1 = k) = P(s_k \mid s, a), \qquad k = 1, \ldots, m.$$

By construction,
$$N(s_k) = \sum_{j=1}^{n} \mathbf{1}\{Y_j = k\}, \qquad \hat{P}(s_k \mid s, a) = \frac{N(s_k)}{n} = \frac{1}{n} \sum_{j=1}^{n} \mathbf{1}\{Y_j = k\}.$$

**Step 1: Empirical and true CDFs.** Define the true CDF
$$F(x) := \Pr(Y_1 \le x), \qquad 1 \le x \le m,$$

and the empirical CDF
$$F_n(x) := \frac{1}{n} \sum_{j=1}^{n} \mathbf{1}\{Y_j \le x\}, \qquad 1 \le x \le m.$$

For integers $k = 1, \ldots, m$,
$$F(k) = \sum_{j=1}^{k} P(s_j \mid s, a), \qquad F_n(k) = \sum_{j=1}^{k} \hat{P}(s_j \mid s, a).$$

**Step 2: DKW inequality.** By the Dvoretzky-Kiefer-Wolfowitz inequality (Dvoretzky et al., 1956), for any $\delta > 0$,
$$\Pr\left( \sup_{1 \le x \le m} \left| F_n(x) - F(x) \right| > \delta \right) \le 2 \exp\left(-2n\delta^2\right). \tag{C.1}$$

**Step 3: From CDF errors to mass-function errors.** For each $k = 1, \ldots, m$ we can write the probability masses using the CDFs:
$$P(s_k \mid s, a) = F(k) - F(k-1), \qquad \hat{P}(s_k \mid s, a) = F_n(k) - F_n(k-1),$$
with the convention $F(0) = F_n(0) = 0$. Therefore
$$\begin{aligned} \left| \hat{P}(s_k \mid s, a) - P(s_k \mid s, a) \right| &= \left| \, [F_n(k) - F_n(k-1)] - [F(k) - F(k-1)] \, \right| \\ &\le \left| F_n(k) - F(k) \right| + \left| F_n(k-1) - F(k-1) \right| \\ &\le 2 \sup_{1 \le x \le m} \left| F_n(x) - F(x) \right|. \end{aligned}$$

Taking the maximum over $k$ yields
$$\max_{1 \le k \le m} \left| \hat{P}(s_k \mid s, a) - P(s_k \mid s, a) \right| \le 2 \sup_{1 \le x \le m} \left| F_n(x) - F(x) \right|.$$

**Step 4: Plug DKW with $\varepsilon/2$.** Let $\varepsilon > 0$. From the previous display we have the inclusion of events
$$\left\{ \max_{1 \le k \le m} \left| \hat{P}(s_k \mid s, a) - P(s_k \mid s, a) \right| > \varepsilon \right\} \subseteq \left\{ \sup_{1 \le x \le m} \left| F_n(x) - F(x) \right| > \frac{\varepsilon}{2} \right\}.$$

Taking probabilities and applying (C.1) with $\delta = \varepsilon/2$ gives
$$\Pr\left( \max_{1 \le k \le m} \left| \hat{P}(s_k \mid s, a) - P(s_k \mid s, a) \right| > \varepsilon \right) \le \Pr\left( \sup_{1 \le x \le m} \left| F_n(x) - F(x) \right| > \frac{\varepsilon}{2} \right) \le 2 \exp\left( -\frac{1}{2} \varepsilon^2 n \right).$$

Replacing $n$ by $N(s, a)$ yields the claimed inequality. $\qquad \square$

### C.4. Induction through the search tree

**Lemma C.5.** *Consider an* VARDE–MCTS *process and a state–action pair* $(s_t, a_t) \in \mathcal{S} \times \mathcal{A}$. *Assume that for every* $s_{t+1} \in \cup_{a \in \mathcal{A}(s_t)} \mathcal{S}(s_t, a)$ *and every fixed* $\varepsilon_{t+1} > 0$, *there exist constants* $C_{s_{t+1}} > 0$ *and* $k_{s_{t+1}}(\varepsilon_{t+1}) > 0$ *such that*

$$\Pr\Big(\big|\hat{V}(s_{t+1}) - V^*(s_{t+1})\big| > \varepsilon_{t+1}\Big) \;\leq\; C_{s_{t+1}} \exp\Big(- k_{s_{t+1}}(\varepsilon_{t+1})\varepsilon_{t+1}^2\, N(s_{t+1})\Big),$$

*then, for every fixed* $\varepsilon > 0$, *there exist constants* $C > 0$ *and* $k_\varepsilon > 0$ *such that*

$$\Pr\Big(\big|\hat{Q}(s_t, a_t) - Q^*(s_t, a_t)\big| > \varepsilon\Big) \;\leq\; C \exp\Big(- k_\varepsilon\, \varepsilon^2\, N(s_t, a_t)\Big).$$

*Proof.* By the assumption, Lemma C.1 and a union bound over $s_{t+1} \in \mathcal{S}(s_t, a_t)$, for every fixed $\varepsilon_1 > 0$ there exist constants $C_1 > 0$ and $k_1(\varepsilon_1) > 0$ such that

$$\Pr(\xi_1) \;\geq\; 1 - C_1 \exp\big(- k_1(\varepsilon_1)\varepsilon_1^2\, N(s_t, a_t)\big),$$

$$\text{with } \xi_1 = \Big\{\forall s_{t+1} \in \mathcal{S}(s_t, a_t): \; \big|\hat{V}(s_{t+1}) - V^*(s_{t+1})\big| \leq \varepsilon_1\Big\}.$$

Moreover, by Lemma C.4, there exist constants $C_2, k_2 > 0$ such that for any $\varepsilon_2 > 0$,

$$\Pr(\xi_2) \;\geq\; 1 - C_2 \exp\big(- k_2\varepsilon_2^2\, N(s_t, a_t)\big),$$

$$\text{with } \xi_2 = \Big\{\max_{s_{t+1} \in \mathcal{S}(s_t, a_t)} \Big|\frac{N(s_{t+1})}{N(s_t, a_t)} - P(s_{t+1} \mid s_t, a_t)\Big| \leq \varepsilon_2\Big\}.$$

Also, by Lemma C.3, there exist constants $C_3, k_3 > 0$ such that for any $\varepsilon_3 > 0$,

$$\Pr(\xi_3) \;\geq\; 1 - C_3 \exp\big(- k_3\varepsilon_3^2\, N(s_t, a_t)\big),$$

$$\text{with } \xi_3 = \Big\{\big|\bar{r}(s_t, a_t) - r(s_t, a_t)\big| \leq \varepsilon_3\Big\}.$$

Under the event $\xi_1 \cap \xi_2 \cap \xi_3$, we have

$$\hat{Q}(s_t, a_t) = \bar{r}(s_t, a_t) + \sum_{s_{t+1} \in \mathcal{S}(s_t, a_t)} \frac{N(s_{t+1})}{N(s_t, a_t)}\, \hat{V}(s_{t+1})$$

$$\leq r(s_t, a_t) + \varepsilon_3 + \sum_{s_{t+1}} \Big(P(s_{t+1} \mid s_t, a_t) + \varepsilon_2\Big)\Big(V^*(s_{t+1}) + \varepsilon_1\Big)$$

$$= r(s_t, a_t) + \sum_{s_{t+1}} P(s_{t+1} \mid s_t, a_t)V^*(s_{t+1}) \;+\; \varepsilon_2 \sum_{s_{t+1}} V^*(s_{t+1})$$

$$+ \; \varepsilon_1 \sum_{s_{t+1}} P(s_{t+1} \mid s_t, a_t) \;+\; \varepsilon_1\varepsilon_2\, |\mathcal{S}(s_t, a_t)| \;+\; \varepsilon_3$$

$$= Q^*(s_t, a_t) \;+\; \varepsilon_2 \sum_{s_{t+1}} V^*(s_{t+1}) \;+\; \varepsilon_1 + \varepsilon_3 \;+\; \varepsilon_1\varepsilon_2\, |\mathcal{S}(s_t, a_t)|.$$

and similarly, by the corresponding lower bounds,

$$\hat{Q}(s_t, a_t) \geq Q^*(s_t, a_t) \;-\; \varepsilon_2 \sum_{s_{t+1}} V^*(s_{t+1}) \;-\; \varepsilon_1 - \varepsilon_3 \;+\; \varepsilon_1\varepsilon_2\, |\mathcal{S}(s_t, a_t)|.$$

For any $\varepsilon > 0$, set

$$\varepsilon_1 = \varepsilon_3 = \frac{\varepsilon}{4}, \qquad \varepsilon_2 = \min\Big\{\frac{\varepsilon}{4\big(1 + \sum_{s_{t+1}} V^*(s_{t+1})\big)}, \frac{1}{|\mathcal{S}(s_t, a_t)|}\Big\}.$$

then give

$$\big|\hat{Q}(s_t, a_t) - Q^*(s_t, a_t)\big| \leq \varepsilon.$$

Since $\xi_1 \cap \xi_2 \cap \xi_3 \to |\hat{Q}(s_t, a_t) - Q^*(s_t, a_t)| \leq \varepsilon$, we have $|\hat{Q}(s_t, a_t) - Q^*(s_t, a_t)| > \varepsilon \to \neg(\xi_1 \cap \xi_2 \cap \xi_3)$. Taking probabilities gives

$$
\begin{aligned}
\Pr\Big(|\hat{Q}(s_t, a_t) - Q^*(s_t, a_t)| > \varepsilon\Big) &\leq \Pr\Big(\neg(\xi_1 \cap \xi_2 \cap \xi_3)\Big) \\
&= \Pr(\neg\xi_1 \cup \neg\xi_2 \cup \neg\xi_3) \\
&\leq \Pr(\neg\xi_1) + \Pr(\neg\xi_2) + \Pr(\neg\xi_3) \\
&\leq C_1 \exp\big(-k_1(\varepsilon_1)\varepsilon_1^2 N(s_t, a_t)\big) \\
&\quad + C_2 \exp\big(-k_2\varepsilon_2^2 N(s_t, a_t)\big) \\
&\quad + C_3 \exp\big(-k_3\varepsilon_3^2 N(s_t, a_t)\big) \\
&\leq C \exp\big(-k_\varepsilon \varepsilon^2 N(s_t, a_t)\big).
\end{aligned}
$$

After fixing $\varepsilon$, the auxiliary tolerances $\varepsilon_1, \varepsilon_2, \varepsilon_3$ are fixed as well. Thus one may take

$$
C = C_1 + C_2 + C_3, \qquad k_\varepsilon = \min\left\{\frac{k_1(\varepsilon/4)}{16}, k_2\left(\frac{\varepsilon_2}{\varepsilon}\right)^2, \frac{k_3}{16}\right\} > 0.
$$

This concludes the proof. $\qquad\square$

**Lemma C.6.** *Consider an* VARDE–MCTS *process. Fix a depth* $t \in \{1, \dots, H\}$ *and a node* $s_t \in \mathcal{S}$. *If for every successor state* $s_{t+1} \in \cup_{a \in \mathcal{A}(s_t)}\mathcal{S}(s_t, a)$ *and every fixed* $\varepsilon_{s_{t+1}} > 0$, *there exist constants* $C_{s_{t+1}} > 0$ *and* $k_{s_{t+1}}(\varepsilon_{s_{t+1}}) > 0$ *such that*

$$
\Pr\Big(|\hat{V}(s_{t+1}) - V^*(s_{t+1})| > \varepsilon_{s_{t+1}}\Big) \leq C_{s_{t+1}} \exp\Big(-k_{s_{t+1}}(\varepsilon_{s_{t+1}})\varepsilon_{s_{t+1}}^2 N(s_{t+1})\Big),
$$

*then, for every fixed* $\varepsilon > 0$, *there exist constants* $C > 0$ *and* $k_\varepsilon > 0$ *such that*

$$
\Pr\Big(|\hat{V}(s_t) - V^*(s_t)| > \varepsilon\Big) \leq C \exp\Big(-k_\varepsilon \varepsilon^2 N(s_t)\Big).
$$

*Proof.* Apply Lemma C.5 to each action $a \in \mathcal{A}(s_t)$ and combine with Lemma C.1 to convert the $N(s_t, a)$-rate into an $N(s_t)$-rate; then union bound over $a \in \mathcal{A}(s_t)$. Concretely, for every fixed $\varepsilon_1 > 0$ there exist constants $C_1 > 0$ and $k_1(\varepsilon_1) > 0$ such that

$$
\Pr(\xi) \geq 1 - C_1 \exp\big(-k_1(\varepsilon_1)\varepsilon_1^2 N(s_t)\big),
$$

$$
\text{with } \xi = \Big\{\forall a \in \mathcal{A}(s_t): |\hat{Q}(s_t, a) - Q^*(s_t, a)| \leq \varepsilon_1\Big\}.
$$

Fix $\varepsilon > 0$ and set $\varepsilon_1 = \varepsilon$. On the event $\xi$,

$$
|\hat{V}(s_t) - V^*(s_t)| = \left|\max_{a \in \mathcal{A}(s_t)} \hat{Q}(s_t, a) - \max_{a \in \mathcal{A}(s_t)} Q^*(s_t, a)\right| \leq \max_{a \in \mathcal{A}(s_t)} |\hat{Q}(s_t, a) - Q^*(s_t, a)| \leq \varepsilon.
$$

Thus $\xi \to \{|\hat{V}(s_t) - V^*(s_t)| \leq \varepsilon\}$, and therefore

$$
\Pr\Big(|\hat{V}(s_t) - V^*(s_t)| > \varepsilon\Big) \leq \Pr(\neg\xi) \leq C_1 \exp\big(-k_1(\varepsilon)\varepsilon^2 N(s_t)\big).
$$

Taking $C = C_1$ and $k_\varepsilon = k_1(\varepsilon)$ concludes the proof. $\qquad\square$

**Theorem C.7** (VARDE–MCTS value estimates). *Consider a* VARDE–MCTS *process. For any depth* $t \in \{0, \dots, H\}$, *any node* $s_t$, *and any fixed tolerance* $\varepsilon > 0$, *there exist constants* $C_{s_t,\varepsilon} > 0$ *and* $k_{s_t,\varepsilon} > 0$ *such that, conditionally on* $N(s_t) = n \geq 1$,

$$
\Pr\Big(|\hat{V}(s_t) - V^*(s_t)| > \varepsilon\Big) \leq C_{s_t,\varepsilon} \exp\big(-k_{s_t,\varepsilon} \varepsilon^2 n\big).
$$

*Moreover, at the root node* $s_0$, *after* $T$ *simulations, there exist constants* $C_\varepsilon > 0$ *and* $k_\varepsilon > 0$ *such that*

$$
\Pr\Big(|\hat{V}(s_0) - V^*(s_0)| > \varepsilon\Big) \leq C_\varepsilon \exp\big(-k_\varepsilon \varepsilon^2 T\big).
$$

*Proof.* Fix $\varepsilon > 0$. The result holds for $t = H + 1$ since $\hat{V}(s_{H+1}) = V^*(s_{H+1}) = 0$. Hence the conditional node-wise bound holds for all depths $t = 0, \ldots, H$ by backward induction using Lemma C.6. At the root, after $T$ simulations we have $N(s_0) = T$, which gives the displayed root bound. Since $\varepsilon > 0$ was arbitrary, the claim follows. $\square$

**Corollary C.8** (VARDE–MCTS root error probability). *Assume the conditions of Theorem C.7 hold. Suppose the optimal root action*

$$a^* = \arg\max_a Q^*(s_0, a)$$

*is unique, and define the optimality gap*

$$\Delta_{\min} := \min_{a \neq a^*} \left( Q^*(s_0, a^*) - Q^*(s_0, a) \right) > 0.$$

*Let $\hat{a}_T = \arg\max_a \hat{Q}_T(s_0, a)$ be the action recommended by VARDE–MCTS after $T$ simulations. Then there exist constants $c, C > 0$ such that*

$$\Pr(\hat{a}_T \neq a^*) \leq C \exp(-cT).$$

*Proof.* By Theorem C.7, for every fixed $\varepsilon > 0$ there exist constants $c_1(\varepsilon), C_1 > 0$ such that

$$\Pr\left( \left| \hat{V}(s_1) - V^*(s_1) \right| > \varepsilon \right) \leq C_1 \exp\left( -c_1(\varepsilon)\varepsilon^2 N(s_1) \right).$$

Lemma C.5 implies that exponential concentration of the value estimates propagates to the root action–value estimates. In particular, for each fixed $\varepsilon > 0$ and each root action $a$, there exist constants $c_2(\varepsilon), C_2 > 0$ such that

$$\Pr\left( \left| \hat{Q}(s_0, a) - Q^*(s_0, a) \right| > \varepsilon \right) \leq C_2 \exp\left( -c_2(\varepsilon)\varepsilon^2 N(s_0, a) \right).$$

Using Lemma C.1, for each fixed $\varepsilon > 0$ there exist constants $c_3(\varepsilon), C_3 > 0$ such that for any root action $a$,

$$\Pr\left( \left| \hat{Q}(s_0, a) - Q^*(s_0, a) \right| > \varepsilon \right) \leq C_3 \exp\left( -c_3(\varepsilon)\varepsilon^2 T \right).$$

Set $\varepsilon = \Delta_{\min}/2$, and define the event

$$\mathcal{E}_T := \bigcap_a \left\{ \left| \hat{Q}(s_0, a) - Q^*(s_0, a) \right| \leq \tfrac{\Delta_{\min}}{2} \right\}.$$

On the event $\mathcal{E}_T$, for any suboptimal action $a \neq a^*$,

$$\hat{Q}(s_0, a^*) \geq Q^*(s_0, a^*) - \tfrac{\Delta_{\min}}{2} > Q^*(s_0, a) + \tfrac{\Delta_{\min}}{2} \geq \hat{Q}(s_0, a),$$

which implies $\hat{a}_T = a^*$.

Therefore,

$$\Pr(\hat{a}_T \neq a^*) \leq \Pr(\mathcal{E}_T^c) \leq \sum_a \Pr\left( \left| \hat{Q}_T(s_0, a) - Q^*(s_0, a) \right| > \tfrac{\Delta_{\min}}{2} \right).$$

Applying the exponential concentration bound from Lemma C.5 and absorbing the finite number of root actions into the constants yields

$$\Pr(\hat{a}_T \neq a^*) \leq C \exp(-cT),$$

which concludes the proof. $\square$

# D. Proofs for Subsection 4.3 (VARDE–Q-LEARNING)

## D.1. Proof of Lemma 4.10 (Coverage)

*Proof.* Fix any state $s \in \mathcal{S}$ and let

$$N_T(s) := \sum_{t=1}^{T} \mathbf{1}\{s_t = s\}, \qquad N_T(s,a) := \sum_{t=1}^{T} \mathbf{1}\{(s_t, a_t) = (s,a)\}.$$

**Step 1 (Min-pulls at each state, VARDE–BAI on the subsequence).** Let $\tau_s(1) < \tau_s(2) < \cdots$ be the (random) times at which $s_t = s$. On this subsequence, define the per-state pull count

$$\widetilde{N}_k^{(s)}(a) := \sum_{j=1}^{k} \mathbf{1}\{a_{\tau_s(j)} = a\}.$$

$$\Rightarrow \widetilde{N}_k^{(s)}(a) = N_{\tau_s(k)}(s,a) \quad \text{and} \quad k = N_{\tau_s(k)}(s).$$

At each visit to $s$, VARDE–Q-LEARNING selects an action by maximizing $\frac{w(s,a)^2 \tilde{\sigma}^2(s,a)}{N(s,a)(N(s,a)+1)}$ (with the convention that the score is $\infty$ when $N(s,a) = 0$), which provides a warm-start $\eta = 1$ on the per-state bandit with arms $\mathcal{A}(s)$. Moreover, since $r \in [0,1]$ and the TD target satisfies $y_t = \hat{r}_t + \gamma \max_b \hat{Q}(s_{t+1}, b) \in [0, H_\gamma]$ with $H_\gamma = \frac{1}{1-\gamma}$, and the $Q$-update is a convex combination of the previous value and $y_t$, we have $\hat{Q}(s,a) \in [0, H_\gamma]$ for all $(s,a)$. Hence the weights satisfy

$$w(s,a) = \frac{e^{\hat{Q}(s,a)/\tau}}{\sum_{b \in \mathcal{A}(s)} e^{\hat{Q}(s,b)/\tau}} = \frac{e^{(\hat{Q}(s,a)/H_\gamma)/(\tau/H_\gamma)}}{\sum_{b \in \mathcal{A}(s)} e^{(\hat{Q}(s,b)/H_\gamma)/(\tau/H_\gamma)}},$$

i.e., the per-state decision rule is exactly VARDE–BAI applied to normalized means $\hat{Q}(\cdot, \cdot)/H_\gamma \in [0,1]$ with temperature $\tau' := \tau/H_\gamma$.

Therefore, applying Lemma B.2 to the per-state bandit (with $K = |\mathcal{A}(s)|$, total pulls $T = k$, warm-start $\eta = 1$, and temperature $\tau'$), for any $k > |\mathcal{A}(s)|$ and any $a \in \mathcal{A}(s)$,

$$\widetilde{N}_k^{(s)}(a) \geq 2e^{-1/\tau'}\bar{\sigma}\left(\frac{k}{|\mathcal{A}(s)|} - \frac{1}{2}\right) - \frac{1}{2} = 2e^{-H_\gamma/\tau}\bar{\sigma}\left(\frac{k}{|\mathcal{A}(s)|} - \frac{1}{2}\right) - \frac{1}{2}.$$

Equivalently, for any time $T$ such that $N_T(s) > |\mathcal{A}(s)|$,

$$N_T(s,a) \geq 2e^{-H_\gamma/\tau}\bar{\sigma}\left(\frac{N_T(s)}{|\mathcal{A}(s)|} - \frac{1}{2}\right) - \frac{1}{2}, \qquad \forall a \in \mathcal{A}(s). \tag{D.1}$$

In particular, if $N_T(s) \to \infty$, then $N_T(s,a) \to \infty$ for every $a \in \mathcal{A}(s)$.

**Step 2 (All states are visited infinitely often).** Let $\mathcal{R}$ be the set of states visited infinitely often:

$$\mathcal{R} := \left\{s \in \mathcal{S} : N_T(s) \to \infty \text{ as } T \to \infty\right\}.$$

Since $|\mathcal{S}| < \infty$ and the process runs forever, $\mathcal{R} \neq \emptyset$ almost surely.

We claim that $\mathcal{R}$ is closed under one-step reachability of any action: if $s \in \mathcal{R}$, $a \in \mathcal{A}(s)$, and $P(s' \mid s,a) > 0$, then $s' \in \mathcal{R}$ almost surely. Indeed, $s \in \mathcal{R}$ implies $N_T(s) \to \infty$, and then (D.1) gives $N_T(s,a) \to \infty$. Let $t_1 < t_2 < \cdots$ be the times when $(s_{t_n}, a_{t_n}) = (s,a)$. At each such time, conditionally on the past, $s_{t_n+1}$ is drawn from $P(\cdot \mid s,a)$, so

$$\Pr\left(s_{t_n+1} = s' \mid \mathcal{F}_{t_n}\right) = P(s' \mid s,a) =: p > 0.$$

Since there are infinitely many such trials with success probability $p > 0$, $s_{t_n+1} = s'$ occurs infinitely often almost surely, which implies $N_T(s') \to \infty$, i.e., $s' \in \mathcal{R}$. Thus, for every $s \in \mathcal{R}$ and every $a \in \mathcal{A}(s)$, all states in the support of $P(\cdot \mid s,a)$ also belong to $\mathcal{R}$. In other words, $\mathcal{R}$ is closed under all actions.

Now take any $s \in \mathcal{R}$ and any $s' \in \mathcal{S}$. By Assumption 4.9, there exists a stationary policy $\pi$ such that $\mathbb{E}_s^\pi[T_{s'}] < \infty$, hence $\Pr_s^\pi(T_{s'} < \infty) = 1$, so there exists some finite $m$ with $\Pr_s^\pi(T_{s'} = m) > 0$. Therefore there exists a length-$m$ path

$$s = s_0 \xrightarrow{a_0} s_1 \xrightarrow{a_1} \cdots \xrightarrow{a_{m-1}} s_m = s'$$

such that $\pi(a_k \mid s_k) > 0$ and $P(s_{k+1} \mid s_k, a_k) > 0$ for all $k$. Since $\mathcal{R}$ is closed under all action transitions and $s_0 \in \mathcal{R}$, it follows inductively that $s_1, \ldots, s_m \in \mathcal{R}$, in particular $s' \in \mathcal{R}$. Because $s' \in \mathcal{S}$ was arbitrary, we conclude $\mathcal{R} = \mathcal{S}$ almost surely, i.e., $N_T(s) \to \infty$ for all $s \in \mathcal{S}$.

**Step 3 (Coverage over state–action pairs).** Finally, combining $N_T(s) \to \infty$ for all $s$ with (D.1) yields $N_T(s, a) \to \infty$ for all $(s, a)$ almost surely, proving the lemma. $\qquad\square$

# E. Full Algorithms

## E.1. VARDE–BAI

---
**Algorithm 1** VARDE–BAI
---

**Parameters:** temperature $\tau$, warm-start $\eta$, variance floor $\bar{\sigma}^2$, budget $T$
**Initialize:** $(N_i, \hat{\mu}_i, \hat{\sigma}_i^2) \leftarrow (0, 0, 0) \ \forall i$
**for** $t = 0$ **to** $K\eta - 1$ **do**
    Pull arm $\tilde{i}_t = t \bmod K$; observe $r_t$; update $N_{\tilde{i}_t}, \hat{\mu}_{\tilde{i}_t}, \hat{\sigma}_{\tilde{i}_t}^2$ by Welford (Alg. 4)
**end for**
**for** $t = K\eta$ **to** $T - 1$ **do**
$$w_i(\hat{\mu}) \leftarrow \frac{\exp(\hat{\mu}_i/\tau)}{\sum_j \exp(\hat{\mu}_j/\tau)}, \ \tilde{\sigma}_i^2 \leftarrow \max\{\hat{\sigma}_i^2, \bar{\sigma}^2\} \ \forall i$$
    Select arm $\tilde{i}_t = \arg\max_i \dfrac{w_i(\hat{\mu})^2 \tilde{\sigma}_i^2}{N_i(N_i + 1)}$; observe $r_t$
    Update $N_{\tilde{i}_t}, \hat{\mu}_{\tilde{i}_t}, \hat{\sigma}_{\tilde{i}_t}^2$ by Welford (Alg. 4)
**end for**
**return** $\hat{i}_T = \arg\max_i \hat{\mu}_i$

---

## E.2. VARDE–MCTS

---
**Algorithm 2** VARDE–MCTS
---

**Input:** root state $s_0$, budget $T$, temperature $\tau$, variance floor $\bar{\sigma}^2$
**Output:** recommended action $\hat{a}_T$

**function** SELECTCHILD($s$)
    **for** $a \in \mathcal{A}(s)$ **do**
$$w(s, a) \leftarrow \frac{\exp(\hat{Q}(s, a)/\tau)}{\sum_b \exp(\hat{Q}(s, b)/\tau)}$$
$$\tilde{\sigma}(s, a) \leftarrow \max\{\hat{\sigma}(s, a), \bar{\sigma}\}$$
    **end for**
    **return** $\arg\max_a \dfrac{w(s, a)^2 \, \tilde{\sigma}^2(s, a)}{N(s, a)(N(s, a) + 1)}$
**end function**

**function** EXPANDSIMULATE($s$)
    Add an unsampled action $a$ to node $s$
    Rollout from $(s, a)$ to depth $H$ to get return $\hat{G}$
    Update $\hat{Q}(s, a) \leftarrow \hat{G}$; $N(s, a) \leftarrow N(s, a) + 1$
    $\hat{V}(s) \leftarrow \max_a \hat{Q}(s, a)$
    **return** $\hat{G}$
**end function**

**function** BACKPROPAGATE(*path*, $\hat{G}$)
    **while** *path* not empty **do**
        Pop last $(s, a, \hat{r})$ from *path*
        $\hat{G} \leftarrow \hat{r} + \gamma\hat{G}$
        Update $\hat{\sigma}^2(s, a)$ from $\hat{G}$ by Welford (Alg. 4)
$$\hat{Q}(s, a) \leftarrow \bar{r}(s, a) + \gamma \sum_{s'} \frac{N(s')}{N(s, a)} \hat{V}(s')$$
        $\hat{V}(s) \leftarrow \max_a \hat{Q}(s, a)$
    **end while**
**end function**

**function** MAINLOOP($s_0$)
    **for** $t = 1, \ldots, T$ **do**
        $s \leftarrow s_0$;    *path* $\leftarrow [\ ]$
        **while** $s$ is not expandable **do**
            $a \leftarrow$ SELECTCHILD($s$)
            Observe $\hat{r} \sim \nu(\cdot | s, a)$
$$\bar{r}(s, a) \leftarrow \frac{N(s, a)\bar{r}(s, a) + \hat{r}}{N(s, a) + 1}$$
            $N(s) \leftarrow N(s) + 1$;    $N(s, a) \leftarrow N(s, a) + 1$
            Transition $s' \sim P(\cdot | s, a)$
            Append $(s, a, \hat{r})$ to *path*; $s \leftarrow s'$
        **end while**
        $\hat{G} \leftarrow$ EXPANDSIMULATE($s$)
        BACKPROPAGATE(*path*, $\hat{G}$)
    **end for**
    **return** $\hat{a}_T = \arg\max_a \hat{Q}(s_0, a)$
**end function**

---

### E.3. VARDE–Q-LEARNING

---

**Algorithm 3** VARDE–Q-LEARNING

---

**Parameters:** budget $T$, discount $\gamma$, temperature $\tau$, variance floor $\bar{\sigma}^2$, initial state $s_0$

**Initialize:** $Q(s,a) \leftarrow \dfrac{1}{1-\gamma}$, $\hat{\sigma}^2(s,a) \leftarrow 0$, $N(s,a) \leftarrow 0 \; \forall (s,a)$; $H_\gamma \leftarrow \dfrac{1}{1-\gamma}$; $s \leftarrow s_0$

**for** $t = 1, \ldots, T$ **do**

$\quad w(s,b) \leftarrow \dfrac{\exp(Q(s,b)/\tau)}{\sum_{c \in \mathcal{A}(s)} \exp(Q(s,c)/\tau)}, \; \forall b \in \mathcal{A}(s)$

$\quad \tilde{\sigma}^2(s,b) \leftarrow \max\{\hat{\sigma}^2(s,b), \bar{\sigma}^2\}, \; \forall b \in \mathcal{A}(s)$

$\quad$ Take action $a \leftarrow \arg\max_{b \in \mathcal{A}(s)} \dfrac{w(s,b)^2 \, \tilde{\sigma}^2(s,b)}{N(s,b)(N(s,b)+1)}$

$\quad$ Observe reward $\hat{r}$ and next state $s'$

$\quad y \leftarrow \hat{r} + \gamma \max_b Q(s',b)$

$\quad$ Update $N(s,a)$, $\hat{\sigma}(s,a)^2$ by Welford on samples of $y$ (Alg. 4)

$\quad \alpha \leftarrow \dfrac{H_\gamma + 1}{H_\gamma + N(s,a)}$; $Q(s,a) \leftarrow (1-\alpha)Q(s,a) + \alpha y$; $s \leftarrow s'$

**end for**

**return** $\hat{\pi}_T(\cdot) = \arg\max_a Q(\cdot, a)$

---

### E.4. Implementation details

**Division by $N(s,a) = 0$.** The VarDE score $\dfrac{w(s,a)^2 \tilde{\sigma}^2(s,a)}{N(s,a)(N(s,a)+1)}$ is set to $+\infty$ when $N(s,a) = 0$. This ensures that all actions are tried at least once before the algorithm starts exploiting the variance estimates.

**Expandable node definition in MCTS.** An expandable node is defined as a state node $s$ where there exists at least one action $a \in \mathcal{A}(s)$ such that $N(s,a) = 0$.

**Welford method for empirical mean and variance update.** We use Welford method to update empirical mean and variance online. The method's pseudo code can be seen in Algorithm 4.

---

**Algorithm 4** Welford update for empirical mean and variance

---

**Input:** count $N \geq 0$, mean $\hat{\mu}$, variance $\hat{\sigma}^2$, new sample $x$

$M_2 \leftarrow N\hat{\sigma}^2$

$N \leftarrow N + 1$

$\delta \leftarrow x - \hat{\mu}$

$\hat{\mu} \leftarrow \hat{\mu} + \delta/N$

$\delta_2 \leftarrow x - \hat{\mu}$

$M_2 \leftarrow M_2 + \delta \, \delta_2$

$\hat{\sigma}^2 \leftarrow M_2/N$

**return** $(N, \hat{\mu}, \hat{\sigma}^2)$

---

# F. Experimental Details

## F.1. BAI Experiments

We evaluate VARDE–BAI on the four following standard Best-Arm Identification (BAI) benchmarks.

- **Experiment BAI.1:** One group of bad arms. Bernoulli distributions. 20 arms. $\mu_0 = 0.2$, $\mu_{1:19} = 0.1$. $T = 1200$.

- **Experiment BAI.2:** Two groups of bad arms. Bernoulli distributions. 20 arms. $\mu_0 = 0.2$, $\mu_{1:5} = 0.12$, $\mu_{6:19} = 0.08$. $T = 1000$.

- **Experiment BAI.3:** Geometric progression. Gaussian distributions. 14 arms. $\mu_0 = 0.4$, $\mu_i = 0.4 - 0.9^{i+9}$ for all $i = 1, \ldots, 13$. $\sigma_i$ is shuffled from $\mu_i$. $T = 200$.

- **Experiment BAI.4:** Ten arms divided in three groups. Gaussian distributions. 10 arms. $\mu_0 = 0.3$, $\mu_{1:3} = 0.22$, $\mu_{4:6} = 0.2$, $\mu_{7:9} = 0.15$. $\sigma_i$ is shuffled from $\mu_i$. $T = 150$.

Besides the error probability at the last pull presented in Table 2, below are additional results showing the evolution of the error probability over time.

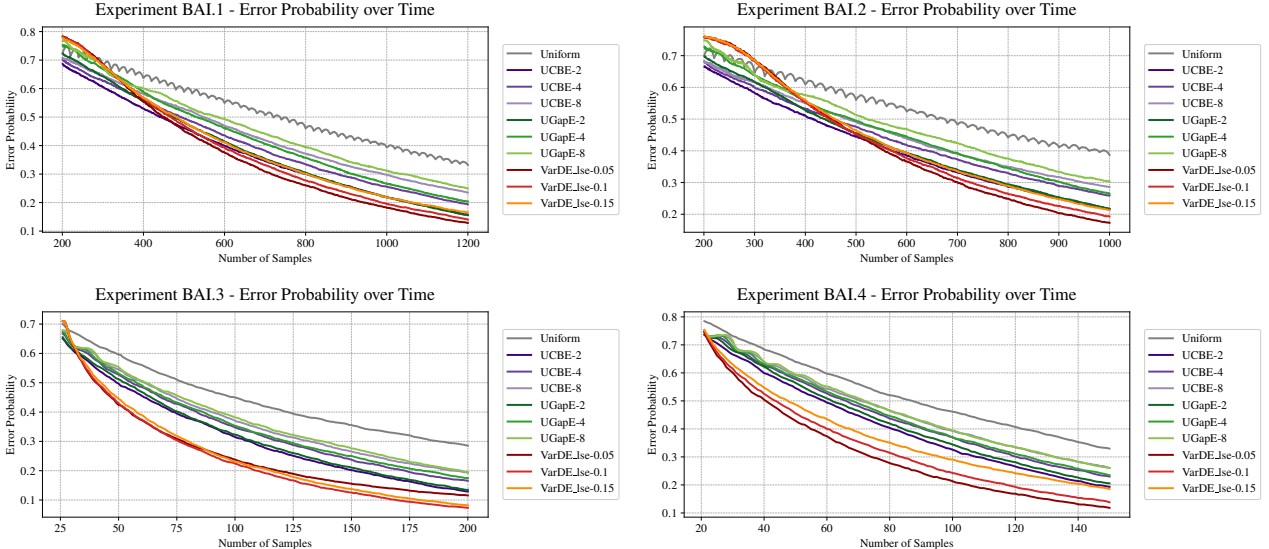

*Figure 5.* Error probability over time for Experiments BAI.1 to BAI.4.

We also provide empirical comparisons of VARDE–BAI with different decision functions on two additional BAI experiments:

- **Experiment BAI.5:** Arithmetic progression. Gaussian distributions. 15 arms. $\mu_i = 0.4 - 0.025i$ for all $i = 0, \ldots, 14$. $\sigma_i$ is shuffled from $\mu_i$. $T = 2000$.

- **Experiment BAI.6:** Two good arms and a large group of bad arms. Gaussian distributions. 20 arms. $\mu_0 = 0.3$, $\mu_1 = 0.28$, $\mu_{2:19} = 0.17$. $\sigma_i$ is shuffled from $\mu_i$. $T = 2000$.

Below are the implemented decision potentials $Y(x)$ and their derivatives $w(x) = \nabla Y(x)$ used by VARDE–BAI. In our experiments, we use $\tau = 0.05$ (LogSumExp), $\mu = 0.5$ (Nesterov), $(\alpha, \mu) = (2, 0.1)$ (EntMax), $\delta = 0.1$ (Pairwise Softplus), and $p = 5$ (Power Mean).

- **LogSumExp (temperature $\tau$):**

$$Y_\tau(x) = \tau \log \Big( \sum_{i=1}^{K} \exp(x_i/\tau) \Big), \qquad w_i(x) = \frac{\exp(x_i/\tau)}{\sum_{j=1}^{K} \exp(x_j/\tau)}.$$

- **Nesterov (quadratic smoothing $\mu$):**

$$Y_\mu(x) = \max_{p \in \Delta_K} \left\{ \langle p, x \rangle - \frac{\mu}{2} \|p\|_2^2 \right\}, \qquad w(x) = \nabla Y_\mu(x) = \Pi_{\Delta_K}(x/\mu),$$

where $\Pi_{\Delta_K}$ denotes Euclidean projection onto the probability simplex.

- **EntMax (Tsallis / $\alpha$-entmax smoothing):**

$$Y_{\alpha,\mu}(x) = \max_{p \in \Delta_K} \left\{ \langle p, x \rangle - \frac{\mu}{\alpha(\alpha-1)} \sum_{i=1}^K p_i^\alpha \right\}, \qquad w(x) = \nabla Y_{\alpha,\mu}(x) = \arg\max_{p \in \Delta_K}(\cdot).$$

Equivalently (as implemented), there exists a threshold $\theta \in \mathbb{R}$ such that

$$w_i(x) \propto \left[ c\,(x_i - \theta) \right]_+^{\frac{1}{\alpha-1}}, \qquad c = \left( \tfrac{\alpha-1}{\mu\alpha} \right)^{\frac{1}{\alpha-1}}, \qquad \sum_{i=1}^K w_i(x) = 1.$$

- **Pairwise Softplus (pairwise smooth-max, parameter $\delta$):** Define, for $a, b \in \mathbb{R}$,

$$m_\delta(a, b) = \frac{a+b}{2} + \sqrt{\left( \frac{a-b}{2} \right)^2 + \delta^2}.$$

Its partial derivatives are

$$\frac{\partial m_\delta}{\partial a}(a, b) = \frac{1}{2} + \frac{a-b}{2\sqrt{\left(\frac{a-b}{2}\right)^2 + \delta^2}}, \qquad \frac{\partial m_\delta}{\partial b}(a, b) = 1 - \frac{\partial m_\delta}{\partial a}(a, b).$$

For $x \in \mathbb{R}^K$, $Y_\delta(x)$ is obtained by repeatedly merging pairs using $m_\delta$ (in a balanced binary-tree order), and $w(x) = \nabla Y_\delta(x)$ is obtained by backpropagating these pairwise derivatives (yielding $w(x) \in \Delta_K$).

- **Power Mean (order $p > 1$):** Let $\tilde{x}_i = x_i$ if $\min_j x_j > 0$, and otherwise $\tilde{x}_i = x_i - \min_j x_j + \varepsilon$ (with $\varepsilon = 10^{-8}$ in code). Define

$$Y_p(x) = \left( \frac{1}{K} \sum_{i=1}^K \tilde{x}_i^p \right)^{1/p}.$$

Its gradient (ignoring the constant shift used to ensure positivity) satisfies

$$\frac{\partial Y_p}{\partial x_i}(x) \propto \tilde{x}_i^{p-1}, \qquad w_i(x) = \frac{\tilde{x}_i^{p-1}}{\sum_{j=1}^K \tilde{x}_j^{p-1}} \quad \text{(simplex-normalized, as implemented).}$$

### F.1.1. COMPONENT ABLATIONS AND PARAMETER SENSITIVITY

We include additional analyses for VARDE–BAI to separate the two factors in the sampling score and to evaluate the sensitivity to the temperature and variance floor. In the component ablation, VAR-ONLY removes the influence weights from the score and uses only the empirical variance term, while WEIGHT-ONLY removes the empirical variance term and uses only the influence weights. Table 3 shows that both components are useful, and the full influence-weighted variance rule performs best at every checkpoint.

*Table 3.* Component ablation for VARDE–BAI: error probability (%) over time.

| Method | 500 | 1000 | 1500 | 2000 |
|---|---|---|---|---|
| Var-Only | 36.40 | 24.83 | 18.22 | 13.09 |
| Weight-Only | 19.29 | 9.37 | 6.40 | 5.07 |
| VarDE | **13.58** | **5.21** | **2.69** | **1.84** |

Table 4 reports error probability over a grid of LSE temperatures $\tau$ and variance floors $\bar{\sigma}^2$. The method is more sensitive to $\tau$ than to the variance floor. The variance floor mainly stabilizes early variance estimates, and performance near the best temperature is stable over a broad range of floors.

*Table 4.* Sensitivity of VARDE–BAI: error probability (%) across temperature $\tau$ and variance floor $\bar{\sigma}^2$.

| $\tau / \bar{\sigma}^2$ | $10^{-4}$ | $10^{-3}$ | $10^{-2}$ | $10^{-1}$ | 1 |
|---|---|---|---|---|---|
| 0.01 | 43.75 | 43.88 | 44.86 | 49.08 | 49.43 |
| 0.03 | 13.55 | 13.22 | 13.57 | 14.78 | 14.94 |
| 0.05 | 6.22 | 5.71 | **5.33** | 5.68 | 5.69 |
| 0.10 | 6.27 | 6.18 | 6.07 | 6.14 | 6.32 |
| 0.20 | 13.15 | 12.88 | 12.87 | 11.28 | 10.93 |
| 0.50 | 19.72 | 19.00 | 19.32 | 17.08 | 16.86 |
| 1.00 | 22.19 | 22.00 | 21.92 | 19.69 | 18.86 |
| 2.00 | 24.16 | 23.76 | 23.47 | 21.11 | 20.16 |

### F.1.2. FIRST-ORDER APPROXIMATION STUDY

To quantify the effect of neglecting higher-order terms, we compare the first-order variance surrogate

$$V_{\text{1st}} = \sum_i w_i(\hat{\mu})^2 \frac{\hat{\sigma}_i^2}{N_i}$$

with the full nonlinear LSE variance

$$V_{\text{full}} = \text{Var}\left[\tau \log \sum_i e^{\tilde{\mu}_i/\tau}\right], \qquad \tilde{\mu}_i \sim \mathcal{N}\left(\hat{\mu}_i, \frac{\hat{\sigma}_i^2}{N_i}\right).$$

Tables 5–7 show that for very small $\tau = 0.05$, the first-order surrogate significantly underestimates the full variance throughout learning. For $\tau = 0.10$, the approximation becomes much tighter as learning progresses. For $\tau = 0.15$, the discrepancy is already moderate early on and quickly becomes very small.

*Table 5.* First-order approximation study for $\tau = 0.05$.

| Step | $V_{\text{full}}$ | $V_{\text{1st}}$ | $V_{\text{full}} - V_{\text{1st}}$ | $\frac{V_{\text{full}} - V_{\text{1st}}}{V_{\text{full}}}$ |
|---|---|---|---|---|
| 20 | $1.227 \times 10^{-2}$ | $3.838 \times 10^{-3}$ | $8.510 \times 10^{-3}$ | 66.4% |
| 400 | $3.659 \times 10^{-3}$ | $2.530 \times 10^{-4}$ | $3.406 \times 10^{-3}$ | 88.7% |
| 600 | $2.790 \times 10^{-3}$ | $1.687 \times 10^{-4}$ | $2.622 \times 10^{-3}$ | 88.0% |
| 900 | $1.916 \times 10^{-3}$ | $1.027 \times 10^{-4}$ | $1.813 \times 10^{-3}$ | 84.9% |

*Table 6.* First-order approximation study for $\tau = 0.10$.

| Step | $V_{\text{full}}$ | $V_{\text{1st}}$ | $V_{\text{full}} - V_{\text{1st}}$ | $\frac{V_{\text{full}} - V_{\text{1st}}}{V_{\text{full}}}$ |
|---|---|---|---|---|
| 20 | $7.112 \times 10^{-3}$ | $2.911 \times 10^{-3}$ | $4.201 \times 10^{-3}$ | 56.1% |
| 400 | $3.609 \times 10^{-4}$ | $2.398 \times 10^{-4}$ | $1.211 \times 10^{-4}$ | 21.6% |
| 600 | $2.157 \times 10^{-4}$ | $1.621 \times 10^{-4}$ | $5.369 \times 10^{-5}$ | 14.4% |
| 900 | $1.198 \times 10^{-4}$ | $1.003 \times 10^{-4}$ | $1.963 \times 10^{-5}$ | 8.4% |

*Table 7.* First-order approximation study for $\tau = 0.15$.

| Step | $V_{\text{full}}$ | $V_{\text{1st}}$ | $V_{\text{full}} - V_{\text{1st}}$ | $\frac{V_{\text{full}} - V_{\text{1st}}}{V_{\text{full}}}$ |
|---|---|---|---|---|
| 20 | $4.644 \times 10^{-3}$ | $2.712 \times 10^{-3}$ | $1.933 \times 10^{-3}$ | 39.5% |
| 400 | $2.540 \times 10^{-4}$ | $2.390 \times 10^{-4}$ | $1.534 \times 10^{-5}$ | 5.3% |
| 600 | $1.682 \times 10^{-4}$ | $1.617 \times 10^{-4}$ | $7.276 \times 10^{-6}$ | 3.8% |
| 900 | $1.026 \times 10^{-4}$ | $1.003 \times 10^{-4}$ | $3.339 \times 10^{-6}$ | 3.0% |

## F.2. MCTS Experiments

### F.2.1. EXPERIMENT MCTS.1 - SAILING ENVIRONMENTS

**Environment.**    The state is $(x, y, d)$, where $(x, y)$ is the agent's position on a $6 \times 6$ grid and $d \in \{0, \dots, 7\}$ is the current wind direction (8 compass directions). From each state, the agent selects one of the 8 compass moves that stays within the grid and is not directly into the wind. After moving, the wind transitions stochastically: it stays the same with probability 0.6, and turns left or right with probability 0.2 each. The reward is dense and negative, corresponding to a movement cost that increases with the angle between the action direction and the wind direction. The agent starts at position $(0, 0)$ and the episode ends when it reaches the goal at position $(5, 5)$ or when the horizon $H = 50$ is reached. The 8 available wind directions and the applicable actions with their rewards under the blue-direction wind are illustrated in Figure 6.

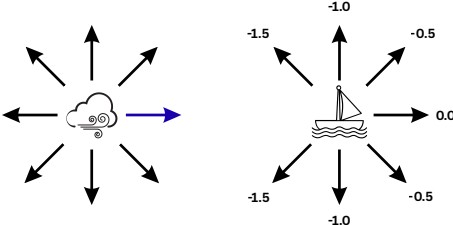

*Figure 6.* Sailing environment wind directions, applicable actions, and rewards under blue-direction wind.

**Evaluation protocol.**    For each algorithm, we run 100 independent runs. Within each run, we perform MCTS simulations and evaluate the policy induced by the current search tree at fixed checkpoints using Monte Carlo rollouts. If a rollout reaches a state that is not represented in the current tree, the evaluation policy defaults to uniform random actions. We evaluate after every 1000 simulations, from 1000 to 30000, and estimate performance using 200 rollouts per checkpoint.

**Hyperparameter selection.**    We select hyperparameters by grid search, using 50 independent runs, 10000 MCTS simulations per run, and 200 evaluation rollouts. For each algorithm, we choose the configuration with the best average Monte Carlo return. The tuned configurations used are: UCT (bias $= 10$, $\varepsilon = 0.1$), MENTS/RENTS/TENTS (temperature $= 0.1$, $\varepsilon = 0.25$), DENTS (temperature $= 2.0$, $\varepsilon = 0.75$), BTS (temperature $= 3.0$, $\varepsilon = 0.25$), and VARDE (temperature $= 1.5$, variance floor $= 100$).

### F.2.2. EXPERIMENT MCTS.2 - TAXI

**Environment.**    Actions are *south*, *north*, *east*, *west*, *pickup*, and *dropoff*. Movement respects the internal wall layout, and in the "raining" setting each move action becomes stochastic: the intended move occurs with probability 0.7, and the two orthogonal moves occur with probability 0.15 each. Rewards follow the standard Taxi design: $-1$ for movement, $-1$ for a successful pickup, $+20$ for a correct dropoff, and $-10$ for illegal pickup/dropoff attempts. We run with horizon $H = 20$, raining enabled.

**Evaluation protocol.**    We evaluate after every 1000 simulations, from 1000 to 30000, using 200 evaluation rollouts per checkpoint, averaged over 100 independent runs.

**Hyperparameter selection.**    We select hyperparameters by grid search, using 50 independent runs, 10000 MCTS simulations per run, and 200 evaluation rollouts. The tuned configurations used are: UCT (bias = auto, $\varepsilon = 0.1$), MENTS (temperature $= 1.0$, $\varepsilon = 0.25$), RENTS (temperature $= 0.1$, $\varepsilon = 0.5$), TENTS (temperature $= 2.0$, $\varepsilon = 0.75$), DENTS (temperature $= 3.0$, $\varepsilon = 0.75$), BTS (temperature $= 5.0$, $\varepsilon = 0.5$), and VARDE (temperature $= 5.0$, variance floor $= 10$).

### F.2.3. EXPERIMENT MCTS.3 - FROZENLAKE

**Environment.**    The state is $(x, y)$, where $(x, y)$ is the agent's position on a $4 \times 4$ map. Each action (`left`, `right`, `up`, `down`) succeeds with probability $p = 0.8$; the remaining probability mass is split equally among the other three directions (slippery environment), with wall/boundary collisions leaving the agent in place. An episode terminates upon reaching the

goal, falling into a hole, or when the horizon $H = 10$ is reached. Rewards are sparse: reaching the goal at time $t$ yields reward $\gamma^t$ (with $\gamma = 0.95$), and all other rewards are 0.

**Evaluation protocol.** We follow the same evaluation procedure as in Experiment MCTS.1. We evaluate after every 1000 simulations, from 1000 to 30000, using 200 evaluation rollouts per checkpoint, averaged over 100 independent runs.

**Hyperparameter selection.** We select hyperparameters by grid search, using 50 independent runs, 10000 MCTS simulations per run, and 200 evaluation rollouts. The tuned configurations used are: UCT (bias = auto, $\varepsilon = 0.2$), MENTS/RENTS (temperature = 0.01, $\varepsilon = 0.85$), TENTS (temperature = 0.05, $\varepsilon = 0.85$), DENTS (temperature = 0.1, $\varepsilon = 0.85$), BTS (temperature = 0.05, $\varepsilon = 0.5$), and VARDE (temperature = 0.015, variance floor = 0.01).

### F.2.4. EXPERIMENT MCTS.4 - SYNTHETIC TREE

**Environment.** We consider a synthetic stochastic tree planning problem. The environment is a balanced $k$-ary tree of depth $d$. At each internal node, the agent selects one of the $k$ outgoing actions, but the transition is noisy: the intended child is taken with probability 0.5 and each other child with probability $0.5/(k-1)$. Episodes terminate upon reaching a leaf, where a terminal reward is sampled as $r \sim \mathcal{N}(\mu_\ell, 0.5^2)$. Leaf means $\mu_\ell$ are generated by sampling i.i.d. edge weights and setting each leaf mean to the sum of weights along its root-to-leaf path, then rescaling all leaf means to $[0, 1]$ within each tree. The optimal root value $V^\star(s_0)$ is computed exactly by dynamic programming under the true transition model. We evaluate across four tree sizes with $(k, d) \in \{(16, 1), (14, 3), (16, 4), (200, 2)\}$.

**Evaluation protocol.** For each $(k, d)$ setting, we generate 5 random trees and run each method for 1000 simulations per tree. We report the value estimation error $|V_t(s_0) - V^\star(s_0)|$ after each simulation $t$, averaged over 5 independent runs per tree (mean $\pm$ 95% CI).

**Hyperparameter selection.** We tune hyperparameters by grid search on a separate validation split (same simulation budget), selecting the configuration that minimizes the average value estimation error. The tuned configurations used are: UCT (exploration coefficient = 0.1), MENTS (exploration coefficient = 0.5, $\tau = 0.1$), RENTS (exploration coefficient = 0.5, $\tau = 0.2$), TENTS (exploration coefficient = 0.5, $\tau = 0.5$), DENTS (exploration coefficient = 0.5, $\tau = 0.1$), BTS (exploration coefficient = 0.25, $\tau = 0.1$), and VARDE (temperature = 0.1, variance floor = 0.001).

### F.3. BPI Experiments

**Environments.** We consider tabular continuing MDPs with Bernoulli rewards $r_t \in \{0, 1\}$. In both environments, episodes start at $s_0 = 0$ and we use discount $\gamma = 0.99$.

RIVERSWIM($L$) has state space $\mathcal{S} = \{0, 1, \ldots, L-1\}$ (so $|\mathcal{S}| = L$) and two actions $\mathcal{A} = \{\texttt{left}, \texttt{right}\}$. The `left` action is deterministic: it moves one step left (and self-loops at $s = 0$). The `right` action is stochastic: for interior states $s \in \{1, \ldots, L-2\}$ it transitions to $s+1$ with prob. 0.3, stays in $s$ with prob. 0.6, and moves to $s-1$ with prob. 0.1; boundary cases follow the standard RiverSwim definition (at $s = 0$, `right` goes to 1 with prob. 0.3 and otherwise stays; at $s = L-1$, `right` goes to $L-2$ with prob. 0.7 and otherwise stays, while `left` goes to $L-2$ deterministically). Rewards are sparse: $\Pr(r_t = 1 \mid s_t = 0, a_t = \texttt{left}) = 0.05$, $\Pr(r_t = 1 \mid s_t = L-1, a_t = \texttt{right}) = 1$, and 0 otherwise.

FORKEDRIVERSWIM($L$) shares a common start state and then splits into two RiverSwim-like branches of length $L$. It has $|\mathcal{S}| = 2L - 1$ states and three actions $\mathcal{A} = \{\texttt{left}, \texttt{right}, \texttt{switch}\}$. Within each branch, `left`/`right` behave as in RIVERSWIM; `switch` deterministically moves to the state at the same depth on the other branch (and self-loops at the start and at the two terminal states). Rewards are again Bernoulli and sparse: $\Pr(r_t = 1 \mid s_t = 0, a_t = \texttt{left}) = 0.05$, the terminal `right` reward is 1 on one branch and 0.95 on the other, and all other rewards are 0.

We run RIVERSWIM with $L \in \{5, 10, 20, 30, 50\}$ and horizons $T \in \{10{,}000; 20{,}000; 40{,}000; 50{,}000; 80{,}000\}$ respectively, and FORKEDRIVERSWIM with $L \in \{3, 5, 10, 15, 25\}$ and horizons $T \in \{20{,}000; 20{,}000; 100{,}000; 200{,}000; 300{,}000\}$ respectively.

**Evaluation protocol.** For each environment size and each method, we run 10 independent runs (seeds $0, \ldots, 9$) for a fixed interaction budget of $T$ steps. During training, we checkpoint every 200 steps and evaluate the greedy policy $\hat{\pi}_t$ induced by the agent. Since the environments are tabular, we evaluate $\hat{\pi}_t$ by exact iterative policy evaluation on the true

transition kernel and expected rewards (tolerance $10^{-6}$), i.e., we do not use Monte Carlo rollouts. We compute $V^*$ by policy iteration on the same model and report the normalized value proximity $1 - \|V^* - V^{\hat{\pi}_T}\|_\infty / \|V^*\|_\infty$ , aggregated as mean $\pm$ 95% CI over the 10 runs.

**Hyperparameter selection.** We use a fixed (singleton) hyperparameter configuration across all environments and sizes. In particular, VARDE–Q-LEARNING uses $(\tau, \texttt{var\_floor}) = (0.1, 0.1)$. Q-UCB uses confidence parameter $\delta = 10^{-3}$. MF-BPI uses $\bar{k} = 1$ and ensemble size 50. MDP-NAS/PS-MDP-NAS use computation frequencies 200 for both the greedy policy and allocation updates, with posterior sampling disabled/enabled respectively. O-BPI uses $\bar{k} = 1$ and update frequency 200.

