# OpenReview forum: "Variance Driven Exploration: A Provable and Efficient Methodology for Pure Exploration in Highly Stochastic Environments"
_ICML.cc/2026/Conference — ICML 2026 regular_

### Official Review · Reviewer_6BSu · 2026-03-11

**Soundness:** 3
**Presentation:** 3
**Significance:** 3
**Originality:** 3
**Overall Recommendation:** 4
**Confidence:** 3

**Summary:**

This paper introduces a decision-variance minimization framework for exploration and integrates it to standard decision task including BAI, MCTS, and best-policy identification. Theoretical guarantees for the convergence of value functions are given. Numerical experiments show the superiority of this framework. The paper is well-written.

**Compliance With Llm Reviewing Policy:**

Affirmed.

**Key Questions For Authors:**

1. Could you clarify the main theoretical challenges introduced by the variance-minimization objective? If its primary effect is to alter the state visiting number, would it be feasible to consider a higher-order approximation of the decision variance under suitable regularity conditions?

2. The paper would benefit from a more principled discussion of how to select the smoothing parameter $\tau$. Since $\tau$ directly governs the bias–smoothness tradeoff, it can materially affect both the theoretical guarantees and the empirical behavior of VarDE. Moreover, the experimental section would be strengthened by a systematic sensitivity analysis over $\tau$, illustrating how performance varies across different problem instances, rather than relying on a fixed (or lightly tuned) choice.

3. Since approximating the max operator typically requires $\tau$ to be small (which in turn can make $M$ large), could you provide numerical evidence on the impact of neglecting higher-order terms in the decision-variance approximation?

**Limitations:**

yes

**Strengths And Weaknesses:**

Strengths
- The paper presents a detailed convergence analysis of the proposed algorithms. The numerical results clearly demonstrate the benefits of variance-driven exploration.

Weaknesses
- From a theoretical perspective, formulating a decision-variance minimization objective appears mainly to influence the visiting number of states without substantially altering standard arguments based on Hoeffding’s inequality(see, e.g. [1]).  In addition, the paper simplifies the decision variance via a first-order approximation, but it remains unclear how this approximation impacts convergence—either in the theoretical analysis (regret) or in the numerical experiments.


Reference

[1]Agarwal, A., Jiang, N., Kakade, S. M., and Sun, W. Reinforcement learning: Theory and algorithms. CS Dept., UW Seattle, WA, USA, Tech. Rep, 32:96, 2019.

---

> ### Author Rebuttal · Authors · 2026-03-31
>
> Thank you for the careful reading and positive assessment. We are glad that the numerical results and convergence analysis came across clearly. We address your main concerns below.
>
> ## (1) Role of the variance-minimization objective vs. standard concentration tools
> We agree that the concentration tool itself is standard, and this is by design. VarDE does not aim to introduce a new Hoeffding-style inequality; its contribution is the **allocation rule**. In fixed-budget pure exploration, the allocation is the algorithm: methods such as UCB-E, UGapE, and Successive Reject all rely on standard concentration arguments, yet differ precisely in how they distribute samples. VarDE changes this distribution by allocating to the component with the largest expected decrease in **decision-level variance**, which is what leads to different finite-budget behavior.
>
> From a theoretical standpoint, this does more than merely perturb visit counts. The variance-minimization objective introduces three nontrivial challenges. First, the influence weights depend on the current estimates and therefore on the past allocation itself, creating an adaptive coupling. Second, in MCTS, local variance reductions must be propagated through stochastic backups to the root recommendation. Third, in Q-learning, one must still prove full coverage of all state-action pairs under a greedy variance-based rule.
>
> ## (2) Feasibility of higher-order approximations
> Yes, this is feasible in principle. A second-order expansion would introduce Hessian-based cross terms of the form
> $$
> (H_f)^2_{ij} \frac{\sigma_i^2\sigma_j^2}{N_i N_j},
> $$
> as well as fourth-moment terms $\kappa_i = \mathbb{E}\left[(X_i-\mu_i)^4\right]$ such as
> $$
> (H_f)_{ii}^2 \frac{\kappa_i}{N_i^3}.
> $$
> These are informative, but they scale as $O(t^{-2})$, whereas the first-order term scales as $O(t^{-1})$. The downside is that using them in practice would require estimating the full Hessian and higher moments online, which is substantially more complex and appears to offer limited benefit in the budget regimes we study. We will clarify this tradeoff in the revision.
>
> ## (3) Choice of the smoothing parameter $\tau$
> We agree this is an important point and a true limitation of the current paper. The choice of $\tau$ governs the bias–smoothness tradeoff of the surrogate LSE, affecting both approximation quality and curvature, and thus the accuracy of the variance decomposition. A more principled or adaptive selection rule for selecting $\tau$ is an important direction for future work.
>
> Empirically, we address this via a sensitivity analysis over $\tau$ (see the response to Reviewer ztPA), showing that performance is stable over a reasonable range and degrades only at extreme values.
>
> ## (4) Impact of the first-order approximation on convergence
> We agree that this needed to be clarified more carefully. In the revised version, we will explicitly state that the first-order approximation is used to derive the **sampling rule**, while the final concentration guarantees are still applied to the **actual empirical means**. Thus, the approximation affects the allocation constants, but not the form of the final guarantees.
>
> More precisely, under the corrected variance decomposition,
> $$
> V_t=\sum_i \frac{w_i(\mu)^2\sigma_i^2}{N_i(t)} + R_t,
> $$
> the remainder satisfies $R_t=O(t^{-2})$, whereas the leading term is $O(t^{-1})$. Therefore, once the estimates have stabilized, the neglected higher-order term is lower order and does not change the variance-decay rate.
>
> For the empirical impact of the approximation, please see the response to Reviewer 7J7s. As shown there, the discrepancy between the first-order surrogate and the full nonlinear variance is largest for very small $\tau$, but becomes much smaller over time for moderate $\tau$. This matches the theoretical bias-curvature tradeoff and supports our claim that the approximation error is controlled.
>
> Overall, we appreciate this feedback. In the revision, we will clarify that VarDE’s novelty lies in the allocation rule, expand the discussion of higher-order approximations, add the new $\tau$ sensitivity analysis and cite the numerical study of the first-order approximation.

---

> > ### Author Rebuttal · Reviewer_6BSu · 2026-04-03
> >
> > Thank you for the response. I will keep my score unchanged.

---

> > > ### Author Response · Authors · 2026-04-08
> > >
> > > We would like to thank Reviewer 6BSu for the constructive discussion and for confirming that our responses adequately addressed the points raised. The feedback regarding the theoretical challenges of variance minimization and the role of the smoothing parameter was particularly insightful. We look forward to incorporating these clarifications into the final manuscript.
> > >
> > > Best regards,\
> > > Authors of Submission 26096

---

### Official Review · Reviewer_7J7s · 2026-03-12

**Soundness:** 3
**Presentation:** 3
**Significance:** 2
**Originality:** 3
**Overall Recommendation:** 4
**Confidence:** 3

**Summary:**

This paper proposes variance driven exploration that allocates sampling effort to directly minimize uncertainty in the final decision output, rather than regulating local estimation errors via optimistic bounds. The key idea is to model the final decision as a smooth function $f(\hat{\mu}_1, \ldots, \hat{\mu}_n)$ of local empirical estimates, then use a first-order Taylor expansion to decompose the decision-level variance which yields a greedy sampling rule. Experiments across all three domains show consistent improvements over baselines, with particularly strong gains in high-variance settings.

This article presents the problem of sample allocation in pure exploration under high stochasticity, where optimism-based methods may systematically misallocate budget.

**Compliance With Llm Reviewing Policy:**

Affirmed.

**Final Justification:**

My major concerns have been adressed suficienty, I maintain my score 4 (weak accep). Reason can be found in myrebuttal acknowledgement.

**Key Questions For Authors:**

Uniform allocation also achieves $\text{Var}[Y_t] = O(t^{-1})$ (since each $N_i = t/n$, the leading term is $\sum_i w_i^2 \sigma_i^2 n / t$). Can you provide a tighter characterization, e.g., the exact leading constant under VarDE versus uniform/UCB-based allocation? Without this, the theorem does not distinguish VarDE from trivial strategies.

**Limitations:**

yes

**Strengths And Weaknesses:**

Strengths:
The core idea, decompose decision-level variance via influence weights, then greedily sample the highest-impact component is intuitive, and broadly applicable. The fact that a single principle yields concrete algorithms for bandits, tree search, and RL is intellectually satisfying and practically appealing.
The influence weight $\times$ variance decomposition (Lemma 3.2) is the conceptual heart of the paper and is well-motivated.

Weaknesses:
Theorem 3.5 is the rate any reasonable allocation strategy achieves (including uniform sampling), since $\text{Var}[\hat{\mu}_i] = \sigma_i^2/N_i$ and $\sum N_i = t$. The $O(t^{-1})$ rate does not capture VarDE's advantage over baselines; a more informative result would characterize the constant in the rate and show it is smaller for VarDE than for uniform or UCB-based allocation.

The first-order approximation has uncontrolled validity early in learning (Soundness). The entire methodology rests on Lemma 3.2, which requires $\hat{\mu} \in B_\epsilon(\mu)$ (Assumption 3.1). Early in learning, when estimates are far from the truth, the quadratic remainder $O(M^2 S^2)$ may dominate the first-order term.

---

> ### Author Rebuttal · Authors · 2026-03-31
>
> Thank you for the careful and constructive review. We appreciate your positive assessment of the decision-variance perspective and your two main concerns. We agree with both points and will revise the paper accordingly.
> ### (1) Theorem 3.5 and the need for a constant-level characterization.
> We agree that the statement $\text{Var}(Y_T)=O(T^{-1})$ is only a coarse rate and does not by itself distinguish VarDE from uniform or other reasonable allocations. The more informative quantity is the leading constant. In the first-order regime, if
> $$N_i(T)=p_iT+O(1),\qquad p_i>0,\quad \sum_i p_i=1,$$
> then
> $$\text{Var}(Y_T) = \sum_{i=1}^n w_i(\mu)^2\frac{\sigma_i^2}{N_i(T)}+R_T = \frac{1}{T}C(p)+O(T^{-2}),$$
> where
> $$C(p):=\sum_{i=1}^n \frac{w_i(\mu)^2\sigma_i^2}{p_i}.$$
> Thus uniform allocation gives $C_{\text{unif}}=n\sum_i w_i(\mu)^2\sigma_i^2$. Minimizing $C(p)$ over the simplex yields
> $$p_i^\star \propto |w_i(\mu)|\sigma_i, \qquad C_\star=\Big(\sum_i |w_i(\mu)|\sigma_i\Big)^2.$$
> By Cauchy-Schwarz, $C_\star\le C(p)$ for any allocation $p$, with equality iff $p=p^\star$. In particular, $C_\star\le C_{\text{unif}}$, and the inequality is strict unless the problem is degenerate.
>
> This is the real advantage of VarDE: not a better $T^{-1}$ exponent, but a better first-order constant. The one-step score $\frac{w_i(\hat\mu)^2\tilde\sigma_i^2}{N_i(N_i+1)}$ is exactly the empirical first-order variance decrement; in the stabilized regime it targets $p^\star$. So VarDE is designed to approach the optimal first-order allocation, while uniform and optimism-based baselines generally converge to different proportions and thus larger constants. We will revise Theorem 3.5 and the discussion to make this explicit.
> ### (2) Soundness of the first-order approximation early in learning.
> We agree that the first-order approximation can be inaccurate early in learning, especially when the curvature of the decision function is large. In our setting, this effect is directly controlled by the smoothing parameter $\tau$. For the LSE decision function used in VarDE-BAI, the Hessian satisfies $\lVert\nabla^2 LSE_\tau\rVert_\mathrm{op} \le \frac{1}{2\tau}$ so smaller $\tau$ increases curvature and amplifies higher-order effects, whereas larger $\tau$ smooths the objective and suppresses them.
>
> To empirically measure the higher-order terms, we ran a dedicated numerical study along the trajectory of the VarDE LSE allocation rule. We compared the first-order approximation $$V_{\mathrm{1st}}=\sum_i w_i(\hat\mu)^2 \frac{\hat\sigma_i^2}{N_i}$$ with the full nonlinear variance $$V_{\mathrm{full}}=\mathrm{Var}\left[\tau \log \sum_i e^{\tilde\mu_i/\tau}\right],$$ where $$\tilde\mu_i \sim \mathcal{N}\left(\hat\mu_i,\frac{\hat\sigma_i^2}{N_i}\right).$$ Thus, $V_{\mathrm{full}}-V_{\mathrm{1st}}$ isolates the higher-order contribution of the nonlinear decision function.
>
> The results show a clear curvature-dependent pattern. For very small $\tau=0.05$, the first-order surrogate significantly underestimates the full variance throughout learning. For $\tau=0.10$, the approximation becomes much tighter as learning progresses. For $\tau=0.15$, the discrepancy is already moderate early on and quickly becomes very small. This supports our claim that the approximation error is not uncontrolled, but explicitly tunable through $\tau$: smaller $\tau$ better approximates the hard max but increases curvature, while moderate $\tau$ yields a substantially more stable variance surrogate. We will add this sensitivity study and clarify this tradeoff in the revision.
>
> **Table 1: $\tau = 0.05$**
> |Step|$V_{\mathrm{full}}$|$V_{\mathrm{1st}}$|$V_{\mathrm{full}}-V_{\mathrm{1st}}$|$\frac{V_{\mathrm{full}}-V_{\mathrm{1st}}}{V_{\mathrm{full}}}$|
> |-:|-:|-:|-:|-:|
> |20|1.227e-02|3.838e-03|8.510e-03|66.4%|
> |400|3.659e-03|2.530e-04|3.406e-03|88.7%|
> |600|2.790e-03|1.687e-04|2.622e-03|88.0%|
> |900|1.916e-03|1.027e-04|1.813e-03|84.9%|
>
> **Table 2: $\tau = 0.10$**
> |Step|$V_{\mathrm{full}}$|$V_{\mathrm{1st}}$|$V_{\mathrm{full}}-V_{\mathrm{1st}}$|$\frac{V_{\mathrm{full}}-V_{\mathrm{1st}}}{V_{\mathrm{full}}}$|
> |-:|-:|-:|-:|-:|
> |20|7.112e-03|2.911e-03|4.201e-03|56.1%|
> |400|3.609e-04|2.398e-04|1.211e-04|21.6%|
> |600|2.157e-04|1.621e-04|5.369e-05|14.4%|
> |900|1.198e-04|1.003e-04|1.963e-05|8.4%|
>
> **Table 3: $\tau = 0.15$**
> |Step|$V_{\mathrm{full}}$|$V_{\mathrm{1st}}$|$V_{\mathrm{full}}-V_{\mathrm{1st}}$|$\frac{V_{\mathrm{full}}-V_{\mathrm{1st}}}{V_{\mathrm{full}}}$|
> |-:|-:|-:|-:|-:|
> |20|4.644e-03|2.712e-03|1.933e-03|39.5%|
> |400|2.540e-04|2.390e-04|1.534e-05|5.3%|
> |600|1.682e-04|1.617e-04|7.276e-06|3.8%|
> |900|1.026e-04|1.003e-04|3.339e-06|3.0%|
>
> Overall, this feedback is very helpful. In the revision we will (i) strengthen Theorem 3.5 by giving the exact first-order constant and comparing it to uniform/baseline allocations, and (ii) clarify the soundness of the first-order approximation by showing how its error is controlled by the smoothing parameter $\tau$, supported by the additional numerical results.

---

> > ### Author Rebuttal · Reviewer_7J7s · 2026-04-03
> >
> > I thank the authors for their thorough response. My primary concerns have been adequately addressed.
> >
> > This paper makes a meaningful contribution to exploration method design. The unified applicability across bandits, tree search, and RL remains a notable strength.
> >
> > However, since the first-order approximation lacks formal validity guarantees during the early stages of learning, I maintain my score. While the authors' clarifications are appreciated, fully resolving this foundational limitation likely extends beyond the scope of a single paper.

---

> > > ### Author Response · Authors · 2026-04-03
> > >
> > > Thank you again for the thoughtful follow-up. We are glad that the broader concerns were largely resolved. We believe the remaining issue is best understood as a gap in the scope of our generic statement, rather than a flaw in the VarDE principle itself. As written, Assumption 3.1 is local: it assumes $f$ is $C^2$ only on $B\_\epsilon(\mu)$, so Lemma 3.1 invokes Taylor’s theorem only under the event $\hat\mu\_t \in B\_\epsilon(\mu)$. Under that wording, your observation is correct.
> > >
> > > However, for the concrete surrogates used in our formal results, this localization issue can be removed altogether. In VarDE-BAI and VarDE-MCTS we use $f=\mathrm{LSE}\_\tau$, and Lemma 4.2 already shows that $|\nabla^2 \mathrm{LSE}\_\tau(x)|\_{\mathrm{op}} \le 1/(2\tau)$ for all $x$, globally. Therefore, for every $t$, Taylor’s theorem gives
> > > $$
> > > f(\hat\mu\_t)=f(\mu)+\nabla f(\mu)^\top(\hat\mu\_t-\mu)+\frac12(\hat\mu\_t-\mu)^\top \nabla^2 f(\xi\_t)(\hat\mu\_t-\mu)
> > > $$
> > > for some $\xi\_t$ on the segment between $\mu$ and $\hat\mu\_t$, and hence
> > > $$
> > > \bigl|f(\hat\mu_t)-f(\mu)-\nabla f(\mu)^\top(\hat\mu_t-\mu)\bigr|
> > > \le \frac{1}{4\tau}|\hat\mu_t-\mu|_2^2.
> > > $$
> > > So the first-order expansion is formally valid at every time step, with an explicit uniform remainder bound, not only after entering a local neighborhood.
> > >
> > > The same point applies state-wise in VarDE-Q-learning: with bounded rewards, the TD target remains bounded, and the Q-update is a convex combination of bounded quantities, so the iterates remain in a bounded domain throughout learning. More generally, if one wants a generic Section 3 statement, it is enough to replace the local assumption by: $f\in C^2$ on a convex domain containing all empirical iterates, with uniformly bounded Hessian on that domain.
> > >
> > > To be precise, this does not claim that the quadratic term is negligible at arbitrarily small budgets. Rather, it shows that the approximation is formally valid uniformly over time, and that its error is explicitly controlled by curvature. When this error becomes small is therefore a quantitative question controlled by the curvature–bias tradeoff induced by $\tau$, which we now clarify in the revision. For this reason, we hope the remaining concern can be viewed as one of theorem scope and presentation in the generic framework, rather than a foundational obstacle to the decision-variance methodology or to the three instantiations studied here.

---

### Official Review · Reviewer_ztPA · 2026-03-13

**Soundness:** 3
**Presentation:** 3
**Significance:** 3
**Originality:** 3
**Overall Recommendation:** 5
**Confidence:** 3

**Summary:**

This article presents the problem of pure exploration under high stochasticity, where the objective is not cumulative reward during learning, but making the best final decision after a fixed sampling budget. The authors argue that common optimistic methods control local uncertainty but do not directly minimize uncertainty of the final returned decision. They propose sampling the component that most reduces a smooth surrogate of decision-level variance, with applications in BAI, MCTS, and BPI.

**Compliance With Llm Reviewing Policy:**

Affirmed.

**Final Justification:**

The authors are encouraged to revise the draft based on the rebuttal.

**Key Questions For Authors:**

Can the authors provide ablations separating influence weights from empirical variance, and some sensitivity analysis for temperature or variance floor?

**Limitations:**

Yes

**Strengths And Weaknesses:**

**Strength**
- The paper is well-motivated. The fact that local confidence bonuses do not necessarily align with the uncertainty of the final decision is indeed a real issue in planning and RL where uncertainty propagates through backups.
- The examples in BAI, MCTS, and tabular Q-learning makes the theories in the proposed exploration principle easier to understand.

**Weaknesses**
- In the proof of Theorem 3.5, the $\Delta R$ term seems to be missing. Would this cause the overall results to differ order-wise?
- The paper should include an exact stylized case study on the theory side to justify the motivations. In particular, a motivating example would help explain why the proposed scheme rule is natural.
- One limitation is that the theoretical result is uneven across applications. the BAI and MCTS sections give finite-sample guarantees, while the Q-learning result is asymptotic convergence. Since the paper is motivated by fixed-budget pure exploration, the RL theory feels weaker than the other parts.

---

> ### Author Rebuttal · Authors · 2026-03-31
>
> Thank you for the positive assessment. We are glad that the motivation resonated and the examples made the principle concrete. We address your main points below.
> ## (1) Missing $\Delta R_t$ in Theorem 3.5
> You are correct that the original appendix proof omitted the $\Delta R_t$ term. We will fix this in the revision. The correction does not change the order of the result.
>
> Let $\Delta_t:=\hat\mu(t)-\mu$ and $g:=\nabla f(\mu)$. A second-order Taylor expansion gives $$Y_t=f(\hat\mu(t))=f(\mu)+g^\top\Delta_t+Q_t,\qquad Q_t=\frac12\Delta_t^\top H_f(\xi_t)\Delta_t.$$ Hence $$V_t=\mathrm{Var}(Y_t)=\sum_{i=1}^n\frac{w_i(\mu)^2\sigma_i^2}{N_i(t)}+R_t,$$ where $$R_t:=\mathrm{Var}(Q_t)+2\mathrm{Cov}(g^\top\Delta_t,Q_t).$$ In the revision, we will present the corrected proof in the following three-step form:
>
> **Leading term.** $$\sum_i\frac{w_i(\mu)^2\sigma_i^2}{N_i(t)}.$$
> **Quadratic remainder.** Since the Hessian is bounded, $|Q_t|\le C\lVert\Delta_t\rVert_2^2$, so
> $$\mathrm{Var}(Q_t)=O(t^{-2}),$$ using bounded observations together with the min-pulls property $N_i(t)=\Omega(t)$.
>
> **Covariance term.** Write $$Q_t=Q_{0,t}+Q_{1,t},\qquad Q_{0,t}:=\frac12\Delta_t^\top H_f(\mu)\Delta_t,$$ where $Q_{1,t}$ is the Lipschitz-Hessian remainder. Independence and centering eliminate the mixed cubic terms in $\mathrm{Cov}(g^\top\Delta_t,Q_{0,t})$, leaving only diagonal third-moment terms, which are $O(t^{-2})$. The remainder satisfies $Q_{1,t}=O(\lVert\Delta_t\rVert_2^3)$, which gives $\mathrm{Cov}(g^\top\Delta_t,Q_{1,t})=O(t^{-2})$ as well.
>
> Therefore $R_t=O(t^{-2})$, and hence $$|\Delta R_t|\le|R_{t+1}|+|R_t|=O(t^{-2}).$$ So the omitted term is strictly lower order.
> ## (2) An exact stylized motivating example
> We thank the reviewer for the suggestion. We note that the paper already includes a motivating example in Figure 1, intended to illustrate the failure mode we target: in a heteroscedastic setting, UGapE under-samples the truly optimal high-variance arm and instead focuses on more stable but suboptimal arms.
>
> We agree, however, that Figure 1 is more of an empirical trajectory than an exact stylized snapshot. In the revision, we will replace it with a simpler 3-arm example. Let Arm 1 be the true best but highly noisy, with $X_1\sim\mathrm{Bernoulli}(0.55)$, and let Arms 2 and 3 be slightly suboptimal but stable, with $X_2\in\{0.48,0.58\}$ and $X_3\in\{0.47,0.57\}$, each with probability $1/2$. Then $\mu_1=0.55>\mu_2=0.53>\mu_3=0.52$, but Arm 1 has much larger variance. Suppose that after 10 pulls per arm, the empirical means are $\hat\mu_1=0.50$, $\hat\mu_2=0.53$, and $\hat\mu_3=0.52$. Although Arm 1 currently looks worst, it is the natural arm to sample next, since it is the only arm whose remaining uncertainty can still overturn the final recommendation.
>
> This example also shows why standard baselines fail. With equal counts, UCB bonuses are the same, so it samples Arm 2 because $\hat\mu_2$ is largest. UGapE similarly keeps refining the Arm 2/Arm 3 comparison instead of revisiting Arm 1. SR is even more brittle: at a phase boundary, it would eliminate Arm 1 because it has the smallest empirical mean. Thus, these methods follow the current empirical ranking, while the only decision-relevant uncertainty is in Arm 1. This is exactly the intuition behind our method: the natural next sample is the arm whose uncertainty can still change the final decision.
> ## (3) Uneven theory across applications
> We agree that this is an important limitation. The BAI and MCTS sections provide finite-sample guarantees, whereas the Q-learning result is currently asymptotic. The RL theorem should be interpreted as a consistency result showing that variance-driven allocation preserves convergence. Establishing non-asymptotic fixed-budget RL guarantees is an important direction for future work.
> ## (4) Ablations and sensitivity
> We agree, and we will include both component ablations and parameter sensitivity.
>
> We compare the error probability (%) when using individual components against the full rule:
> |Step|500|1000|1500|2000|
> |-|-:|-:|-:|-:|
> |Var-Only|36.40|24.83|18.22|13.09|
> |Weight-Only|19.29|9.37|6.40|5.07|
> |VarDE|13.58|5.21|2.69|1.84|
>
> Both components matter: VarDE < Weight-Only < Var-Only; combining them yields the strongest performance.
>
> We also analyze error probability across different values of the temperature parameter ($\tau$) and the variance floor ($\bar\sigma$):
> |$\tau$ \ $\bar\sigma$|0.0001|0.001|0.01|0.1|1.0|
> |-|-:|-:|-:|-:|-:|
> |0.01|43.75|43.88|44.86|49.08|49.43|
> |0.03|13.55|13.22|13.57|14.78|14.94|
> |0.05|6.22|5.71|5.33|5.68|5.69|
> |0.1|6.27|6.18|6.07|6.14|6.32|
> |0.2|13.15|12.88|12.87|11.28|10.93|
> |0.5|19.72|19.00|19.32|17.08|16.86|
> |1.0|22.19|22.00|21.92|19.69|18.86|
> |2.0|24.16|23.76|23.47|21.11|20.16|
>
> The method is more sensitive to $\tau$ than to $\bar\sigma$. The variance floor mainly stabilizes early estimates, performance near the best $\tau$ is comparatively stable across a broad range of $\bar\sigma$.

---

> > ### Author Rebuttal · Reviewer_ztPA · 2026-04-04
> >
> > The authors are encouraged to revise the draft based on the rebuttal.

---

> > > ### Author Response · Authors · 2026-04-08
> > >
> > > We sincerely thank Reviewer ztPA for the positive final assessment and for acknowledging that the concerns have been fully resolved. Your suggestions regarding the stylized motivating example and the ablations/sensitivity analysis have significantly strengthened the paper. We will ensure all these improvements are polished in the final camera-ready version.
> > >
> > > Best regards,\
> > > The Authors of Submission 26096

---

### Decision · Program_Chairs · 2026-04-30

**Decision:**

Accept (regular)

**Comment:**

This paper proposes variance driven exploration that allocates sampling effort to directly minimize uncertainty in the final decision output. The key idea is to model the final decision as a smooth function of local empirical estimates, then use a first-order Taylor expansion to decompose the decision-level variance which yields a greedy sampling rule. Experiments across three domains show consistent improvements over baselines, with particularly strong gains in high-variance settings.

The paper is well-motivated. The core idea of this paper, decomposing decision-level variance via influence weights, then greedily sampling the highest-impact component, is intuitive, and broadly applicable. The numerical results in this paper clearly demonstrate the benefits of variance-driven exploration.

All reviewers recommend accepting this paper. After reading the paper, the reviews, and the rebuttals, I agree with the reviewers and also recommend accepting this paper.